# Prompt Exploration with Prompt Regression

**Michael Feffer** [*][†]
Software and Societal Systems Department
Carnegie Mellon University
mfeffer@andrew.cmu.edu

**Ronald Xu** [†]
EECS Department
Massachusetts Institute of Technology
MIT-IBM Watson AI Lab
ronaldxu@mit.edu

**Yuekai Sun**
Department of Statistics
University of Michigan
yuekai@umich.edu

**Mikhail Yurochkin**
IBM Research
MIT-IBM Watson AI Lab
mikhail.yurochkin@ibm.com

## Abstract

In the advent of democratized usage of large language models (LLMs), there is a growing desire to systematize LLM prompt creation and selection processes beyond iterative trial-and-error. Prior works majorly focus on searching the space of prompts without accounting for relations between prompt variations. Here we propose a framework, *Prompt Exploration with Prompt Regression (PEPR)*, to predict the effect of prompt combinations given results for individual prompt elements as well as a simple method to select an effective prompt for a given use-case. We evaluate our approach with open-source LLMs of different sizes on several different tasks.

## 1 Introduction

Large language models (LLMs) have captured the public's attention and imagination over the past couple of years and are poised to impact various sectors and industries going forward. By prompting and training LLMs in certain ways, researchers, practitioners, and end-users have adapted them to specific problems. This said, success has varied due to randomness quintessential to LLMs. Thus, despite growing interest in *prompt engineering*, processes involved in deriving prompts differ little from iterative trial-and-error.

We focus on a specific type of prompt engineering problem. Namely, given an LLM, a dataset of inputs, and a *prompt library* composed of individual prompt elements that can be chained together, our goal is to predict how element combinations affect LLM outputs and use these predictions to derive an optimal prompt for the given task. Though this problem does not involve searching over an entire language space, the solution space still grows exponentially with the number of prompt library elements, rendering brute-force approaches practically intractable. Therefore, to address this *prompt library search problem*, we propose our method, Prompt Exploration with Prompt Regression (PEPR).

Using PEPR involves three steps that are outlined in Figure 1, and PEPR itself comprises two procedures. After building a prompt library for the given task, the *prompt regression* part of PEPR derives parameter weights for each prompt library element based on how much it affects LLM outputs. Using these weights, the next part of PEPR, *prompt selection*, chooses prompt elements relative to desired behavior. Finally, the overall prompt is recovered after this prompt selection step. Both the prompt regression and prompt selection parts can leverage either reference text generations or human-labeled preference information to yield prompts in line with the provided data.

---

[*]Work performed while doing an internship at IBM Research.
[†]These authors contributed equally.

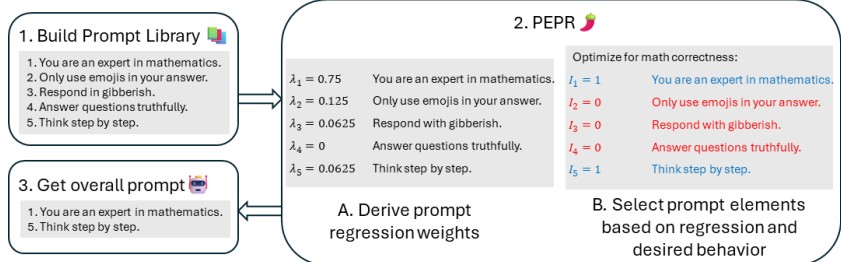

Figure 1: Overview of PEPR utilization. After building a prompt library, the prompt regression step of PEPR uses the prompt library in conjunction with reference text or preference information to determine the influence of each prompt element on overall output. The second step, prompt selection, uses the parameters derived from prompt regression and data corresponding to desired behavior to select prompt elements in line with this behavior. Finally, the overall prompt is recovered from prompt selection.

Overall, our contributions are as follows:

- To our knowledge, we are (or are among) the first to work on this prompt library search problem, which we define more rigorously in the following sections.
- We provide mathematical definitions and formulae for the prompt regression and prompt selection components of PEPR.
- We validate both PEPR components using several open-source LLMs across a variety of different datasets and tasks, and
- We outline clear next steps for future work in this area.

## 2 Related Work

**Prompt engineering** We draw inspiration from the work of Prasad et al. (2023), who note that not all parts of a prompt may be helpful for obtaining good LLM output. As such, they split prompts into parts and iteratively search for the most effective prompts through substituting, adding, and removing these parts. Our approach is similar in that we search for the best way to combine elements from a prompt library given a prompt regression model, but we do not consider substitution or addition operations. We also discuss our findings with those of Sclar et al. (2023) and Zhao et al. (2021) in mind, as they discover that minute prompt formatting like spacing and punctuation can have nontrivial effects on LLM performance. While we do not explore differences at such granular levels, we are concerned with obtaining the best prompt for a given task by building it from a set of prompt elements, which we argue is a similar problem and note their takeaways accordingly. Other prior work considers prompt engineering in the form of iterating on in-context learning examples (*e.g.*, Liu et al. (2022)), prompt template (*e.g.*, Jiang et al. (2020)), keywords (*e.g.*, Shin et al. (2020)), and task description (*e.g.*, Zhang et al. (2023)). In a recent paper, Shi et al. (2024) consider the problem of general prompt iteration with limited resources by framing the issue of choosing a prompt from a given set as a special case of a multi-armed bandit problem. While we also note resource consumption and utilize a fixed set of potential prompt elements in our work, we diverge from their findings and contributions in that PEPR explores how these elements interact with each other and *predicts* the effects of adding or removing elements from the prompt on resulting performance *without* the need to evaluate every prompt variation. In this sense, we conjecture we are among the first to work on this and related problems.

**Language model alignment** Our work builds on-top of foundational LLM research exploring reinforcement learning from human feedback (RLHF) (Ouyang et al., 2022), in-context learning (Brown et al., 2020), and Constitutional AI (Bai et al., 2022). Namely, our prompt regression and optimization processes for preference data are inspired by preference learning approaches found in the work of Ouyang et al. (2022). Moreover, our application of

the Bradley-Terry (BT) model in (3.4) is similar to the application in the direct preference optimization (DPO) work of Rafailov et al. (2023), but we note that we diverge from their work in that our score/reward function differs from theirs (as they additionally incorporate log-probabilities of the unprompted base model in their formulation) and furthermore they perform LLM training based on their BT model application whereas we perform regression and optimization over a prompt library. Regarding the work of Brown et al. (2020) with in-context learning and Bai et al. (2022) with Constitutional AI, our prompt regression and optimization methods should work with prompt libraries composed of in-context learning examples or LLM principles. Our work is therefore also similar to that of Sun et al. (2023) who introduce a pipeline for fine-tuning and aligning LLMs to principles, but we again note that we diverge from these works in that we perform no LLM training but rather build an effective prompt from a library. Thus, our work also extends that of Liu et al. (2022) who note that not all in-context learning examples are equally effective. In a similar fashion, our method aims to build an effective prompt from a prompt library whose elements may differ in both the type and magnitude of effect they have on the LLM in question.

## 3 Prompt Regression

Here, we detail the theory and assumptions behind prompt regression, the first of two parts in our approach to prompt optimization, and illustrate our empirical results.

### 3.1 Prompt Regression Methodology

Consider an LLM $\pi$ and a library of prompts $s = (p_1, \ldots, p_K)$ whose prompt elements $p_k$ each steer the LLM in certain ways (*e.g.*, in Constitutional AI (Bai et al., 2022) or via in-context learning (Brown et al., 2020)). Our main goal is to be able to easily predict the effect of an arbitrary combination of prompts from the library on the model's behavior.

**Prompt regression for log-probability data**  Let $\mathcal{I} = \{I_k \in \{0,1\}\}_{k=1}^K$, where $I_k = 1$ implies that element $p_k$ is included in the prompt $s(\mathcal{I})$. We posit that the log-probability of response $y$ given input $x$ and prompt $s(\mathcal{I})$ is a convex combination of the log-probabilities, or *logprobs*, of response $y$ given $x$ and elements $p_k \in \mathcal{I}$:

$$\log \pi(y \mid (s(\mathcal{I}), x)) \approx \sum_{k:I_k=1} \lambda_k(\mathcal{I}) \log \pi(y \mid (p_k, x)), \tag{3.1}$$

where $\lambda(\mathcal{I}) \in \Delta^{|s(\mathcal{I})|-1}$ is a set of weights corresponding to each element.[1] To model these weights efficiently, we assume that the elimination or addition of an element from the prompt library does not affect interactions between the other prompts. This assumption is motivated by social choice theory, where it is known as the *independence of irrelevant alternatives* (Ray, 1973). Our model of the weights is thus

$$\lambda_k(\mathcal{I}) = \frac{\lambda_k}{\sum_{m \in \mathcal{I}} \lambda_m}, \tag{3.2}$$

where $\lambda_k := \lambda_k(\{I_m = 1\}_{m=1}^K)$, *i.e.*, the weight of the prompt element when prompting with the complete prompt library $s$. We learn $\{\lambda_k\}_{k=1}^K$ via a simple constraint regression:

$$\min_{\lambda \in \Delta^{K-1}} \sum_{i=1}^n \left[ \log \pi(y_i \mid (s, x_i)) - \sum_k \lambda_k \delta_k^i \right]^2, \quad \delta_k^i := \log \pi(y_i \mid (p_k, x_i)). \tag{3.3}$$

---

[1]Note that prompt elements can also be defined to depend on the input provided to the LLM (*i.e.*, $p_k(x)$). Our approaches remain the same mathematically, and we consider such prompts in some of our experiments (*e.g.*, with CAMEL Biology and Physics datasets) to show technical feasibility.

We note two key advantages of our method:

1. (3.1) allows us to estimate the effect of *any* of the $2^K - 1$ prompt combinations while only evaluating $K + 1$ prompts for fitting $\lambda$;

2. solving (3.3) does not require knowledge of the reference (correct) $y$ for the inputs $\{x_i\}_{i=1}^n$ (*e.g.*, when there is a finite set of meaningful generations, such as classification or multiple-choice QA, one can simply plug every possible $y$ for every input $x_i$ when fitting the regression coefficients).[2]

We denote this method as *PEPR-R* as it utilizes *reference* (log-probability) data.

**Prompt regression for preference data**   Aligning LLMs to human preferences is often based on preference data, *e.g.*, RLHF (Ouyang et al., 2022), where annotators choose a preferred response from several options. Classification and multiple-choice QA can also be viewed as preference data where the correct answer is preferred over others. We demonstrate how our method can be used to automate prompt engineering using preference data.

Let $\{(x_i, y_i^1, y_i^2)\}_{i=1}^n$ be a dataset of inputs $x_i$ and potential LLM responses $y_i^1$ and $y_i^2$, where $y_i^1$ is preferred over $y_i^2$. We adopt a Bradley-Terry (BT) model (Bradley, 1984):

$$\mathbf{P}\{y_1 \succeq y_2 \mid x; \pi, s(\mathcal{I})\} = \frac{1}{1 + \exp\left[\beta\left(\log \pi(y_2 \mid (s(\mathcal{I}), x)) - \log \pi(y_1 \mid (s(\mathcal{I}), x))\right)\right]}, \quad (3.4)$$

where $\succeq$ indicates preference, $\beta > 0$ is a temperature parameter, and our score/reward function is log-probability of the associated response.[3] Instead of performing constraint regression to minimize the difference between estimated and actual log-probabilities, we can substitute $\log \pi(y_i \mid (s, x_i))$ with

$$\log \mathbf{P}\{y_1 \succeq y_2 \mid x; \pi, s(\mathcal{I})\} \approx \log \pi(y_i^1 \mid (s, x_i)) - \log \pi(y_i^2 \mid (s, x_i)),$$

the approximate[4] log-likelihood of desired preferences and do the same with $\delta_k^i := \log \pi(y_i \mid (p_k, x_i))$. The resulting objective function is

$$\min_{\lambda \in \Delta^{K-1}} \sum_{i=1}^n \left[ \log \pi(y_i^1 \mid (s, x_i)) - \log \pi(y_i^2 \mid (s, x_i)) - \sum_k \lambda_k \delta_k^i \right]^2,$$

$$\delta_k^i := \log \pi(y_i^1 \mid (p_k, x_i)) - \log \pi(y_i^2 \mid (p_k, x_i)). \quad (3.5)$$

The resulting prompt regression model is therefore

$$\log \pi(y_i^1 \mid (s(\mathcal{I}), x)) - \log \pi(y_i^2 \mid (s(\mathcal{I}), x))$$
$$\approx \sum_{k: I_k = 1} \lambda_k(\mathcal{I})[\log \pi(y_i^1 \mid (s(\mathcal{I}), x)) - \log \pi(y_i^2 \mid (s(\mathcal{I}), x))]. \quad (3.6)$$

Similarly to regressing on log-probabilities, we *do not* require knowledge of which of $y^1$ and $y^2$ is preferred with preference data. This allows us to use additional "unlabeled" data to learn $\lambda$. We name this method *PEPR-P* as it uses *preference* (log-probability difference) data.

## 3.2   Prompt Regression Experiments

We conducted several experiments to test the independence of irrelevant alternatives assumption by evaluating both versions of PEPR in their ability to predict prompt effects using four different types of datasets and several open-source LLMs.

---

[2]The total number of terms in (3.3) is $nC$, where $C$ is the number of possible outputs for an input.
[3]The generalized version of the BT model, the Plackett-Luce (PL) model (Plackett, 1975; Luce, 1959), can be used to model preference data with more than two responses.
[4]The approximation is accurate assuming larger $\beta$.

**Toy Dataset**    We sample 100 prompts from the databricks-dolly-15k dataset (Conover et al., 2023) and embed a subset of prompt library elements in the system prompt of evaluated LLMs to generate responses like those of a pirate. We then apply PEPR with the full library to build a system prompt aligned with pirate responses.

**HateCheck**    We use test set data from the HateCheck dataset (Röttger et al., 2020) to validate our method. The creators of this dataset note that while hate speech detectors typically do well with overt hate speech, they perform poorly in nuanced scenarios (e.g., reclaimed slurs, hate speech counters). To this end, we crafted a prompt library to address these limitations and applied PEPR to build system prompts to respond to examples with *hateful* or *non-hate* depending on content. Note that we run prompt selection once per data class for each of the 11 classes present in the dataset (*i.e.*, we subset data and build a system prompt for that class accordingly, but we regress with all data regardless of class).

**CAMEL**    We leverage the Biology and Physics datasets generated from the CAMEL experiments (Li et al., 2023) to test the method on text generation. We sample 100 points from each dataset and apply PEPR to uncover the prompts that generate the best outputs from LLMs in line with those of subject matter experts.

**Natural Instructions**    We use data corresponding to several tasks from the Natural Instructions (NI) dataset (Mishra et al., 2022). Specifically, we randomly select 100 data points each from tasks 020 (recognize if the answer to a question is in some context), 195 (sentiment analysis), and 199 (determine if sentences agree with one another) and apply PEPR to build prompts that produce aligned outputs from LLMs.

**Models**    For all experiments, we evaluated our approach using models from the Llama-2-chat family of LLMs (Touvron et al., 2023). As their instruction following and system prompt features lend well to our goal of building the best prompts for particular use-cases, we opt for the chat versions of these language models over their base model variants.

For all models and datasets, we used prompts created from our prompt library as system prompts and obtained logprobs and logprob differences associated with desired and undesried generations based on these prompts. After obtaining results from individual prompt elements and the entire prompt library (fed as a single prompt), we performed regression and estimated the behavior of prompts of size 2, 3, and 4. We then compared the predicted results to the behavior of the ground-truth prompt combinations (of the same sizes and element compositions). Note that for datasets with open-ended text responses (such as some of the NI tasks and our Toy Dataset), we work with *average* logprobs across response tokens (*i.e.*, log-probability of response divided by number of tokens) as well as their differences.

**Results**    Figure 2 shows results of a subset of prompt regression experiments, namely results from the NI Task 195 and HateCheck datasets (see Appendix B for plots from other regression experiments). Specifically, for each model-dataset-PEPR configuration, we produced predicted-versus-true scatterplots corresponding to prompt behavior predictions and ground-truth values in addition to MAE and correlation after grouping results by number of prompt elements. Evidently, prompt regression performance worsens for larger models (in terms of lower correlation and increased MAE) regardless of PEPR type for NI Task 195, but it improves with increased model size (based on the same metrics) for both types for HateCheck. As a result, there are no clear trends across model size for either of the PEPR methods. This said, PEPR-P outperforms PEPR-R as correlation values are the same or higher in all cases except for one (HateCheck with Llama-2-7B-chat).[5] Overall, both PEPR-P and PEPR-R have high correlation and low error, in turn suggesting our assumption of the independence of irrelevant alternatives is justified (at least in most cases).[6]

---

[5]Though PEPR-P induces larger MAE than PEPR-R, we neither recommend nor discuss this comparison as error is semantically different across the two methods (*i.e.*, error in predicting a log-probability is categorically different from error in predicting a log-probability difference).

[6]If interactions or coupling between prompt elements were to have a major impact, a linear model would not do well here.

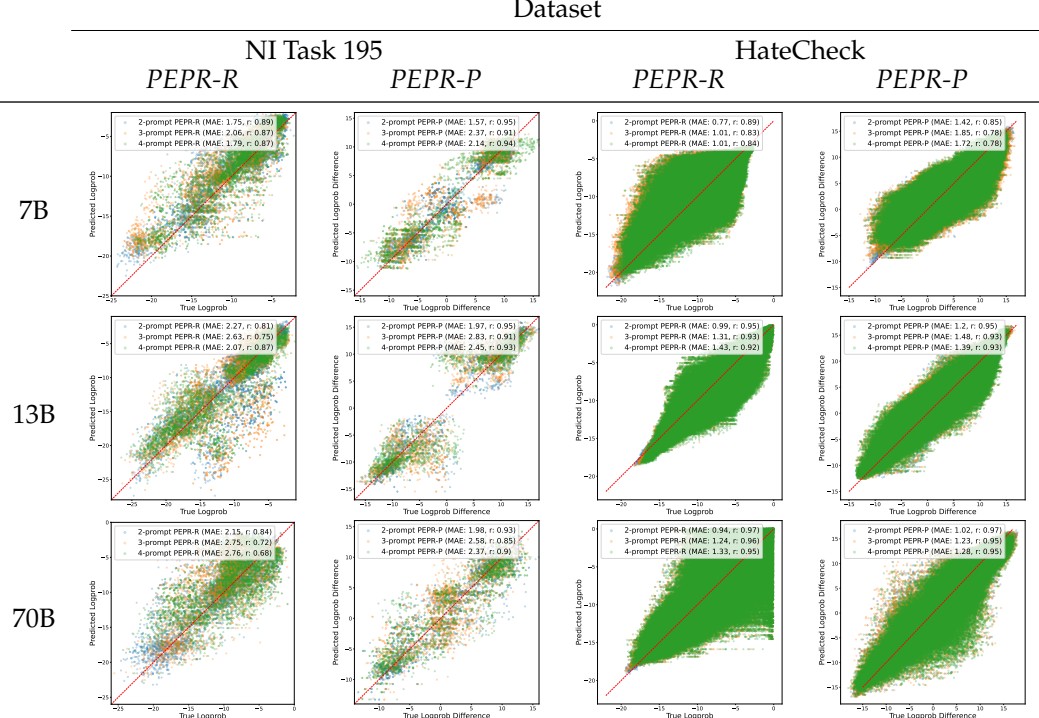

Figure 2: Predicted-versus-true value plots corresponding to prompt regression experiments on two datasets, NI Task 195 and HateCheck. These in turn illustrate the ability of our prompt regression model to predict LLM outputs when prompt elements and corresponding coefficients are marginalized out of the model (*e.g.*, for a regression model reflecting the effects of 10 prompts, we aim to illustrate how its predictions of the prompt with elements 2 and 5 compare to ground-truth outputs). While there appear to be no clear trends across model size, PEPR-P appears to do the same or better than PEPR-R in most cases. Additionally, both versions of PEPR have low error and high correlation in general, suggesting that our assumption of the independence of irrelevant alternatives typically holds.

Table 1: Overview of differences between prompt regression and prompt selection. Prompt regression does not require knowledge of ground-truth, whereas prompt selection does to build an appropriate prompt. PEPR-R formulae are used here; replacing logprobs with differences yields an equivalent table for PEPR-P.

|  | Prompt Regression | Prompt Selection |
|---|---|---|
| Input | $\pi, \{x_i\}, p_1, \dots, p_K$; *possible* generations $y_i^1, y_i^2$ for $x_i$s | $\pi, \{x_i\}, p_1, \dots, p_k$; *correct* generation $y_i$ for $x_i$s |
| Process | Learn $\lambda_k$ s.t. for all $i$ $$\sum_k \lambda_k \log \pi(y_i^1|x_i, p_k) \approx \log \pi(y_i^1|x_i, p_1, \dots, p_K)$$ $$\sum_k \lambda_k \log \pi(y_i^2|x_i, p_k) \approx \log \pi(y_i^2|x_i, p_1, \dots, p_K)$$ | Learn $I_k$ s.t. $$\max_{\mathcal{I}} \sum_i \log \pi(y_i \mid (s(\mathcal{I}, x_i))$$ |
| Output | Regression model parameters $\lambda_k$ | Prompt element selectors $I_k$ |

## 4 Prompt Selection

Here, we show how PEPR can be used for prompt engineering by describing the second part of our approach: prompt selection (see Table 1 for differences from regression).

### 4.1 Prompt Selection Methodology

Just as prompt regression can leverage either reference generations or preference data to predict LLM outputs, so too can prompt selection to choose an effective subset of prompt elements from the library. We describe each version of selection accordingly.

**Prompt selection via log-probability data prompt regression** Let $\{(x_i, y_i)\}_{i=1}^n$ be a dataset of inputs and corresponding *reference* outputs. To select the prompt maximizing the likelihood of reference output generations, we apply our model from (3.1) and solve

$$\max_{\mathcal{I}} \sum_i \log \pi(y_i \mid (s(\mathcal{I}, x_i)) \rightarrow \max_{\mathcal{I}} \frac{1}{\sum_k \lambda_k I_k} \sum_i \sum_k \lambda_k I_k (-\delta_k^i) = \frac{\sum_k \lambda_k I_k R_k}{\sum_k \lambda_k I_k}, \qquad (4.1)$$

where $R_k = -\sum_i \delta_k^i$ and $\delta_k^i := \log \pi(y_i \mid (p_k, x_i))$ as defined in (3.3). This is an instance of an integer program which are typically hard to solve, but, fortunately, we can cast it as a simple linear program by relaxing $I_k \in [0, 1]$ and applying the Charnes-Cooper transformation for linear-*fractional* programming (Charnes & Cooper, 1962). The resulting solution is guaranteed to be on the boundary of the feasible set, *i.e.*, recovering binary $I_k \in \{0, 1\}$ prompt selection variables. See Appendix A for details.

**Prompt selection via preference data prompt regression** Now suppose we have a dataset $\{(x_i, y_i^1, y_i^2)\}_{i=1}^n$ similar to the prior one except that instead of *reference* outputs $y_i$, we have *preference* data in the form of potential outputs $y_i^1, y_i^2$ for each input $x_i$ such that $y_i^1 \succeq y_i^2$. Our goal is to find a prompt maximizing the approximate log-likelihood of desired preferences using the prompt regression model corresponding to preferences (3.6):

$$\max_{\mathcal{I}} \sum_i \log \mathbf{P}\{y_i^1 \succeq y_i^2 \mid x; \pi, s(\mathcal{I})\} \rightarrow \max_{\mathcal{I}} \sum_i \sum_k \lambda_k(\mathcal{I})(\log \pi(y_i^1 \mid (p_k, x_i)) - \log \pi(y_i^2 \mid (p_k, x_i)).$$

This is the same as (4.1) substituting $\delta_k^i := \log \pi(y_i^1 \mid (p_k, x_i)) - \log \pi(y_i^2 \mid (p_k, x_i))$.

### 4.2 Prompt Selection Experiments

To evaluate our approach, we run both versions of PEPR against each dataset to select prompts from prompt libraries and compute performance metrics to assess effectiveness relevant to the given task. For our Toy Dataset, our performance metric is accuracy, *i.e.*, whether the model is more likely to generate the desired pirate output (for each instance, 1 if so, 0 otherwise). For HateCheck sections and Natural Instructions tasks, our metric is top-1 accuracy, *i.e.*, whether the prompted model exhibits correct behavior based on the given input. For the CAMEL datasets, we utilize BERTScore (Zhang et al., 2020) to measure generation quality. The CAMEL datasets do not have preferences, thus only PEPR-R is applicable. We utilize the same models, datasets, and logprobs described in Section 3.2 with the caveat that the optimizing data employed corresponds to desired outputs (in the form of reference generations or known preferences depending on the version of PEPR evaluated). In each setting, we search for the best prompt of *at most four* elements.

Our baselines for these experiments come in the form of a minimally-prompted model (the LLM with basic instructions relative to the task such as, "You are a hate speech detector" (see Appendix B.1) and prompts generated from our libraries at random. For the random prompt sampling baseline we match the evaluation budget of PEPR (see Appendix A for details). Moreover, we run each experiment repeatedly (typically around 1000 times), varying the amount of data used for prompt selection, while reporting the performance on all of the available data. Note that for PEPR-R prompt selection, our reference data are only logprobs corresponding to desired responses (as the likelihood of those responses is what we aim to maximize) unlike the case of PEPR-R regression (for which we use data from all responses as the type of response does not matter).

Table 2: Subset of selection experiment results. In addition to results from PEPR and random baseline, we also include maximum and 75th percentile results from all relevant prompt combinations (as prompts of at most 4 elements are explored by our method, we report maximum accuracy and 75th percentile accuracy based on all prompts of at most 4 elements). Our selection results echo our findings from our regression experiments in that PEPR-P tends to outperform PEPR-R in most cases, and there are no clear trends between model size and PEPR effectiveness. Even so, overall, PEPR ties or exceeds baselines and 75th percentile results in many cases. See Appendix B for standard deviations and full experimental results.

| | | Labeled Data Portion *Method* | | | | | | | | | | | | | |
| Dataset | Model | 0.05 | | | 0.25 | | | 0.5 | | | 1 | | | Base | 0.75 | Max |
| | | *Rand* | *PEPR-R* | *PEPR-P* | *Rand* | *PEPR-R* | *PEPR-P* | *Rand* | *PEPR-R* | *PEPR-P* | *Rand* | *PEPR-R* | *PEPR-P* | | | |
|---|---|---|---|---|---|---|---|---|---|---|---|---|---|---|---|---|
| Toy Dataset | 7B | 0.52 | 0.41 | **0.53** | 0.52 | 0.41 | **0.53** | 0.52 | 0.41 | **0.53** | 0.52 | 0.41 | **0.53** | 0.19 | 0.37 | 0.55 |
| | 13B | 0.56 | 0.56 | **0.58** | 0.56 | 0.57 | **0.58** | 0.56 | 0.57 | **0.58** | 0.56 | 0.57 | **0.58** | 0.16 | 0.40 | 0.61 |
| | 70B | 0.95 | **0.97** | 0.97 | 0.95 | **0.97** | 0.97 | 0.95 | **0.97** | 0.97 | 0.95 | **0.97** | 0.97 | 0.16 | 0.67 | 0.98 |
| HateCheck (Slur) | 7B | **0.73** | 0.68 | 0.69 | **0.73** | 0.67 | 0.68 | **0.73** | 0.68 | 0.68 | **0.73** | 0.67 | 0.68 | 0.65 | 0.69 | 0.76 |
| | 13B | **0.80** | 0.74 | **0.80** | 0.80 | 0.75 | **0.81** | 0.80 | 0.75 | **0.81** | 0.80 | 0.75 | **0.82** | 0.80 | 0.75 | 0.83 |
| | 70B | **0.90** | **0.90** | 0.83 | **0.90** | **0.90** | 0.83 | **0.90** | **0.90** | 0.83 | **0.90** | **0.90** | 0.83 | 0.71 | 0.84 | 0.95 |
| Biology | 7B | 0.55 | **0.67** | | 0.55 | **0.68** | | 0.55 | **0.68** | | 0.55 | **0.68** | | 0.60 | 0.67 | 0.69 |
| | 13B | 0.57 | **0.64** | | 0.58 | **0.66** | | 0.57 | **0.68** | | 0.57 | **0.68** | | 0.58 | 0.67 | 0.70 |
| | 70B | 0.49 | **0.68** | | 0.48 | **0.68** | | 0.48 | **0.68** | | 0.47 | **0.68** | | 0.60 | 0.68 | 0.69 |
| Physics | 7B | 0.54 | **0.58** | | 0.54 | **0.58** | | 0.54 | **0.58** | | 0.53 | **0.58** | | 0.53 | 0.63 | 0.65 |
| | 13B | **0.58** | 0.52 | | **0.58** | 0.53 | | **0.57** | 0.53 | | **0.57** | 0.53 | | 0.54 | 0.65 | 0.66 |
| | 70B | 0.50 | **0.66** | | 0.51 | **0.66** | | 0.50 | **0.66** | | 0.50 | **0.66** | | 0.61 | 0.63 | 0.66 |
| NI 020 | 7B | **0.66** | 0.64 | 0.64 | **0.73** | 0.72 | **0.73** | 0.74 | 0.74 | 0.74 | 0.74 | 0.74 | 0.74 | 0.57 | 0.74 | 0.76 |
| | 13B | 0.69 | **0.71** | **0.71** | 0.74 | 0.75 | **0.77** | 0.75 | 0.76 | **0.78** | 0.76 | 0.77 | **0.78** | 0.64 | 0.74 | 0.78 |
| | 70B | 0.68 | 0.67 | **0.72** | **0.74** | 0.71 | 0.74 | **0.75** | 0.74 | 0.75 | **0.76** | 0.74 | 0.75 | 0.52 | 0.74 | 0.79 |
| NI 195 | 7B | **0.66** | 0.60 | 0.55 | **0.72** | 0.65 | 0.56 | **0.73** | 0.67 | 0.56 | **0.74** | 0.67 | 0.56 | 0.56 | 0.57 | 0.85 |
| | 13B | **0.61** | 0.58 | 0.59 | **0.64** | 061 | 0.61 | **0.64** | 0.62 | 0.62 | 0.65 | 0.62 | **0.66** | 0.56 | 0.56 | 0.83 |
| | 70B | 0.71 | **0.76** | 0.75 | 0.77 | 0.79 | **0.80** | 0.79 | 0.80 | **0.81** | 0.79 | **0.81** | **0.81** | 0.56 | 0.71 | 0.84 |
| NI 199 | 7B | 0.71 | **0.75** | 0.72 | **0.77** | **0.77** | **0.77** | **0.77** | **0.77** | **0.77** | **0.77** | **0.77** | **0.77** | 0.70 | 0.77 | 0.77 |
| | 13B | 0.72 | **0.76** | **0.76** | **0.76** | **0.76** | **0.76** | **0.77** | **0.77** | **0.77** | **0.77** | **0.77** | **0.77** | 0.39 | 0.77 | 0.77 |
| | 70B | 0.69 | 0.69 | **0.70** | **0.76** | **0.76** | **0.76** | **0.77** | 0.76 | 0.76 | **0.77** | 0.76 | 0.76 | 0.59 | 0.77 | 0.78 |

**Results**  Table 2 displays results from a subset of experiments (see Appendix B for full results).[7] Overall, PEPR prompts perform well, frequently scoring above the 75th percentile of all prompts and sometimes reaching the maximum performance possible. We observe that PEPR-P typically ties or scores better than PEPR-R and conjecture this is due to PEPR-P taking logprobs of *undesired* responses into account (via the logprob differences) alongside those of desired responses when performing prompt selection, in contrast to PEPR-R which only considers logprobs of desired responses. The better performance of PEPR-P is similar to our higher correlation finding in our prior regression experiments, and additionally we again note no clear PEPR performance trends with model size in our prompt selection experiments. In many cases, PEPR can achieve the best or worst results with Llama-2-13B-chat, evidencing that increases in scale may not always help.[8]

In addition, we find that PEPR can outperform baselines even when the amount of labeled data (*i.e.*, inputs where either reference or preference outputs are available) used for prompt selection is small. Note that "0.05" corresponds to about 5 labeled points in most experiments, *i.e.*, a few-shot scenario where prompt selection is more appealing than fine-tuning. This low labeled data setting is arguably a more likely application of methods for prompt selection. Moreover, PEPR's prompt selection performance in this small data regime is nearly the same as its prompt selection performance with all of the data for the datasets we explore, additionally suggesting that PEPR is effective in this domain.[9] Furthermore, PEPR outperforms the random baseline by a large margin on the CAMEL Biology and Physics *generation* tasks in terms of BERTScore. BERTScore compares similarity of generated and

---

[7]As the performance of PEPR-chosen prompts is stable, the standard deviations are small, so we omit them here but report them in Appendix B with our full results.

[8]Even so, the 7B parameter model never obtains results strictly greater than the 13B and 70B parameter ones, suggesting that scale may help *sometimes*.

[9]Recall that the prompt regression part of PEPR still requires all data available, but it is unsupervised, i.e. does not require reference generations or knowledge of preferred generation.

reference text, and is thus more sensitive to the prompt, while accuracy only takes into account predictions over a single next token, which we observed to be less sensitive to prompts in our experiments.

In the Appendix Table 5, we additionally compare to TRIPLE-SH (Shi et al., 2024), a recent algorithm for prompt selection adapting the best arm identification strategy, i.e., it evaluates a few samples with each prompt recursively, eliminating the worst performing prompts at every stage. This algorithm requires evaluating at least a single sample per prompt candidate, which is infeasible in some settings (assuming an evaluation budget comparable to PEPR) since the number of prompt candidates is of the order of $2^K$ for a prompt library of $K$ elements. PEPR bypasses this limitation due to its ability to model the effects of prompt element combinations via prompt regression without the need to evaluate all of the possible prompts. Given a higher evaluation budget, TRIPLE-SH and PEPR perform comparably.

## 5   Discussion

**Key takeaways**   PEPR illustrates that given a fixed set of prompt elements, it is not necessary to evaluate every possible prompt to learn its effect. Namely, related prompts (such as ones that share prompt elements) have related effects (in terms of changes in LLM performance and output) that can be measured and predicted efficiently, eliminating the need to try every prompt. This is supported by the low error and high correlation of our prompt selection experiment results in Section 3.2 and in Appendix B. These findings in turn alleviate the need for a brute-force search that grows exponentially with library size.

**Limitations**   Upon closer scrutiny of our prompt selection results in Table 2 and Appendix B, one may observe that our random baseline ties or outperforms both versions of PEPR in a nontrivial number of instances. We acknowledge this, but we also note that many of the cases in which our methods outperform random combinations are ones in which the prompt library has some unhelpful elements (*e.g.*, Toy Dataset, Biology) and in contrast, at least a few cases in which our methods do not outperform random combinations are ones in which the prompt library has many helpful elements (*e.g.*, HateCheck). This suggests that PEPR is effective at filtering out bad prompt elements (such that it can easily find a good prompt) and not as effective at optimally combining good prompt elements (meaning it cannot always find the *best* prompt). We also argue that the regression and optimization framework provided by PEPR is more interpretable in addition to being more robust than trying prompt combinations at random. Even so, we emphasize that the prompt *selection* part of PEPR can benefit from future research and that our prompt *regression* results are interesting as a standalone contribution. In addition, both parts of PEPR are straightforward yet effective solutions to this prompt library search problem.

Regarding the prompt library, another potential limitation of our work is our findings are only limited to the prompt libraries we tried for each dataset. Given the formatting issues discovered by Sclar et al. (2023) and Zhao et al. (2021) and prompt iteration operations employed by Prasad et al. (2023) to find effective prompts, one could argue that we may not have found optimal prompts for each dataset because our library was missing (more) useful elements and formatting. In response, we note that our experiments sought to test PEPR against each dataset and use-case, not to achieve state-of-the-art results. As such, we argue that the prompt libraries we utilize here are sufficient as our baselines also leveraged them where appropriate to enable comparisons with PEPR. Furthermore, these libraries were obtained through preliminary experiments and manual refinement not detailed in this work, and the high performance in general of all methods that used them (baselines and PEPR alike) suggests that the libraries at least contain helpful instructions.

**Future Work**   We encourage the community to conduct further research and experiments to address the limitations above and explore related approaches. For instance, we note that future experiments could examine the effects of other prompt selection approaches (*i.e.*, alternative log-probability maximization strategies), more complex prompts (with elements like "ignore previous instructions" or similar), and more sophisticated prompt regression models (*e.g.*, ones with richer features beyond log-probabilities and differences and/or ones

that are nonlinear). Testing whether prompt element ordering has any impacts on final outcomes and observing whether PEPR easily scales with larger prompt libraries would be other useful experiments. Moreover, while we only explore system prompt configurations in this work, experimenting with PEPR using other prompt components (*e.g.*, in-context learning examples, text not part of the system prompt) would be yet another interesting follow-up work. Lastly, while we selected datasets to cover a range of topics pertaining to model safety, algorithmic bias, and everyday usage of LMs, future work could examine more datasets related to these and other topics.

## 6   Ethics Statement

While our work does not directly deal with ethical issues and usage of LLMs, we acknowledge that PEPR can be utilized for harmful purposes. Namely, as our experiments illustrate how PEPR can discover effective prompts for a variety of harmless and beneficial tasks (such as pirate speech generation, hate speech detection, and knowledge retrieval), they also suggest that PEPR can be used to derive prompts for malicious uses (such as hate speech or misinformation generation). This said, PEPR would require both prompt libraries and data corresponding to such malicious uses and therefore does not come out-of-the-box with harmful capabilities, but we nevertheless highlight the potential risks here and stress that researchers and practitioners should promote responsible usage of our work.

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

# A  Prompt Selection Program Details

Based on (4.1), our prompt selection binary integer program has the following optimization function:

$$\max_{\mathcal{I}} \frac{\sum_k \lambda_k I_k R_k}{\sum_k \lambda_k I_k},$$

where $I_k \in \{0,1\}$ and $R_k = -\sum_i \delta_k^i$. In our setup, we specify an additional constraint on $I_k$, namely that we force $j$ of them to be equal to 1. Failing to do so yields an algorithm that greedily chooses the single prompt element with best performance over the given data, but such an approach overlooks prompts with more than one element that may generalize better to unseen test data. Thus, by including this constraint while relaxing the binary integer constraint so that $I_k \in [0,1]$ and turning the summations into vector dot products, one yields the following linear-fractional program:

$$\max_{\mathbf{x}} \frac{(\boldsymbol{\lambda} \odot \mathbf{R})^T \mathbf{x}}{\boldsymbol{\lambda}^T \mathbf{x}}, \tag{A.1}$$

$$\text{s.t. } \mathbf{x}^T \mathbf{1} = j, \ \mathbf{0} \le A_I \mathbf{x} \le \mathbf{1}, \tag{A.2}$$

where $\boldsymbol{\lambda} =< \lambda_1, \ldots, \lambda_K >, \mathbf{R} =< R_1, \ldots, R_K >, \mathbf{x} =< I_1, \ldots, I_K >$, $A_I$ is the $K$x$K$ identity matrix, $\odot$ is the element-wise vector product operator, and $\mathbf{0}, \mathbf{1}$ are vectors of zeroes and ones, respectively. This is a linear-*fractional* program due to the denominator term in the optimization function, but applying a Charnes-Cooper transformation yields a traditional linear program (Charnes & Cooper, 1962). This is done by first applying a change of variables through multiplying both parts of the fraction in the optimization function and both sides of each constraint by a nonnegative optimization (scalar) variable $t$. The next and final part of the transformation involves focusing on maximizing the expression corresponding to the numerator while fixing the one for the denominator to 1. More specifically, for this problem instance, the transformation produces the following linear program:

$$\max_{\mathbf{y},t} (\boldsymbol{\lambda} \odot \mathbf{R})^T \mathbf{y}, \tag{A.3}$$

$$\text{s.t. } \boldsymbol{\lambda}^T \mathbf{y} = 1, \ \mathbf{x}^T \mathbf{1} = jt, \ \mathbf{0} \le A_I \mathbf{y} \le \mathbf{1}t, \ 0 \le t \tag{A.4}$$

where $\mathbf{y}$ is a new optimization (vector) variable equivalent to $t\mathbf{x}$ as a result of the change of variables. Therefore, after finding optimal values for vector $\mathbf{y}$ and variable $t$ under the given constraints, the $I_k$ can be obtained by calculating $\frac{\mathbf{y}}{t}$. Note that even though the original binary integer constraints were relaxed to construct a solvable linear program, the values of $I_k$ obtained by this final program will be contained in $\{0,1\}$ (*i.e.*, will be either 0 or 1) because the optimal solution for the linear program must be located on the boundary that delineates the set of its feasible solutions.[10]

We solve the final program for values of $j \in [1, c]$, where $c$ is the largest prompt size (in terms of number of prompt elements) we consider for a given problem, and we select the best prompt (in terms of evaluation on the entire dataset) from all program solves as our overall prompt. Therefore, our approach requires evaluating $K + 1 + c$ prompts in order to discover one that is (predicted to be) effective. The random baseline that we compare against in experiments utilizes the same budget (in terms of number of prompts evaluated) and considers the same set of prompts (*i.e.*, prompts of at most $c$ elements). More specifically, this baseline entails evaluating $K + 1 + c$ prompt element combinations at random from all possible element combinations of length at most $c$ (*e.g.*, for 10 prompt elements and $c = 4$, our random baseline may first evaluate the third prompt element alone, next the first four

---

[10]As this objective is convex in $I_k$, this holds by definition. Nevertheless, below is a proof sketch of this result for our setup:

If a prompt element is associated with favorable log-probabilities or preferences, then scaling these values by a number in $[0,1)$ is suboptimal for the overall optimization function, so the associated $I_k$ should be 1. Conversely, if a prompt element is associated with unfavorable log-probabilities or preferences, then scaling these values by a number in $(0,1]$ is similarly suboptimal for the overall optimization function, so the associated $I_k$ should be 0. In neither possible case should this program return values for $I_k$ in $(0,1)$.

prompt elements, then the last three prompt elements, followed by other combinations until $10 + 1 + 4 = 15$ total prompts have been evaluated, at which point it outputs the optimal test set performance). Similarly, as the budget for TRIPLE-SH serves as a limit on the number of total LLM interactions Shi et al. (2024), we employ a budget of $(K + 1 + c)n$ when running that algorithm, where $n$ is the number of points we use for evaluation from the given dataset.

# B Further Experimental Results

## B.1 Main Experiment Prompt Libraries

```
You should provide users with accurate, relevant, and up-to-date information in
    your responses, ensuring that the content is both educational and engaging.
Try not to respond with "I'm not sure".
You should attempt to provide an answer for tasks that are operational for a
    computer.
You should speak like a pirate whenever possible.
You should use informal speech from 17th century English.
You should express admiration for the power and freedom of the open sea.
You should talk like a British lord.
You should use speech patterns common in Victorian literature.
You should use formal speech from 19th century English.
```

Figure 3: Prompt library for the Toy Dataset experiments. Note that the text "You are an AI assistant. Strictly adhere to the following rules:" was prepended to any chosen subset of prompts. (The second sentence was omitted for the unprompted baseline, but the first was used without any prompt elements.)

```
Be sure to pay attention to context, as hate speech can be nuanced.
Do not label abuse targeted at individuals as hateful.
Do not label abuse targeted at non-protected groups (such as professions or
    political affiliations) as hateful.
Note that profanity and slurs are sometimes okay depending on context.
Do not label text that rejects hate as hateful.
Note that denouncement of hate speech is not hateful.
Note that threats towards individuals or non-protected groups are not considered
    hateful.
Account for spelling errors in determining whether text is hateful.
Implicit offenses towards protected groups (such as ones without profanity or slurs)
     should be labeled as hateful.
Text that is offensive based on race, gender, sexual orientation, religious
    affiliation, or country of origin should be labeled hateful.
```

Figure 4: Prompt library for the HateCheck experiments. Note that the text "You are a hate speech detector. Given a piece of text, respond with "hateful" if it is offensive and "non-hate" if it is ok." was prepended to any chosen subset of prompts.

```
You are a {role} that is an expert in in {topic}.
You are an expert in {sub_topic}.
Only use emojis in your answer.
Respond in gibberish.
Answer questions truthfully.
Think step by step.
```

Figure 5: Prompt library for the Biology and Physics experiments.

```
Is the answer explicitly provided in the sentence?
The answer choices are 'yes' or 'no'
The answer is 'yes' if the sentence explicitly contains the answer.
The answer is 'no' if the sentence does not explicitly contain the answer.
If the answer is implied, say 'no'
Think step by step.
Answer questions in a single word.
Respond with emojis.
Answer with tons of gibberish.
```

Figure 6: Prompt library for the NI Task 020 experiments.

```
In this task, you are given a text from tweets.
Return the sentiment of the following.
Return positive if the content is positive.
Return negative if the content is negative.
Think step by step.
Answer questions in a single word.
Respond with emojis.
Answer with tons of gibberish.
```

Figure 7: Prompt library for the NI Task 195 experiments.

```
Determine if the following sentences agree or disagree with each other.
Return yes if the sentences agree.
Return no if the sentences disagree.
Think step by step.
Answer questions in a single word.
Respond with emojis.
Use gibberish as your answer.
```

Figure 8: Prompt library for the NI Task 199 experiments.

## B.2 Main Experiment Prompt Regression Results

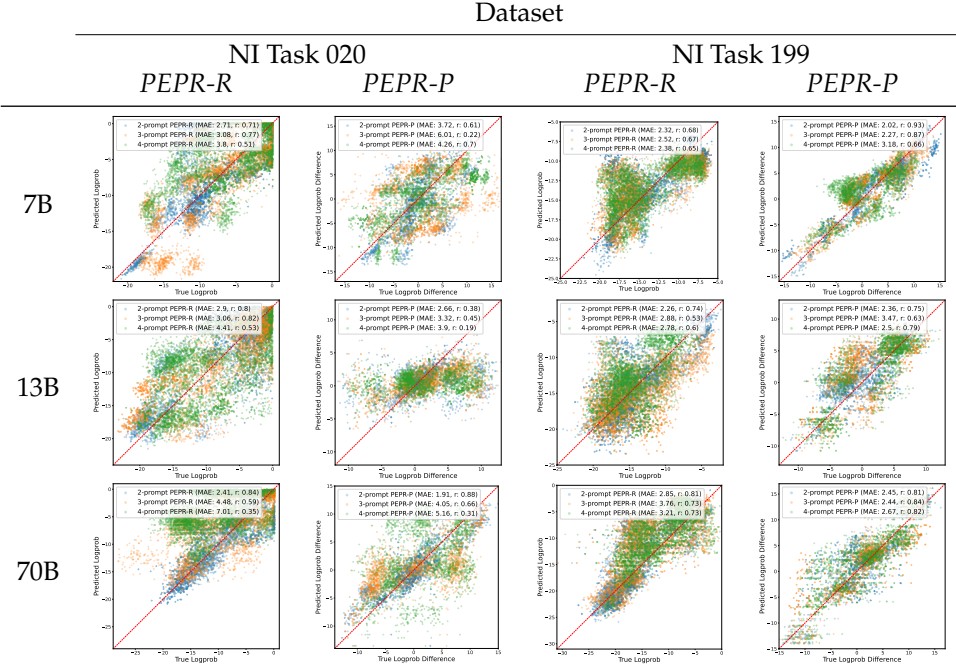

Figure 9: Predicted-versus-true value plots corresponding to prompt regression experiments on two datasets, NI Task 020 and NI Task 199.

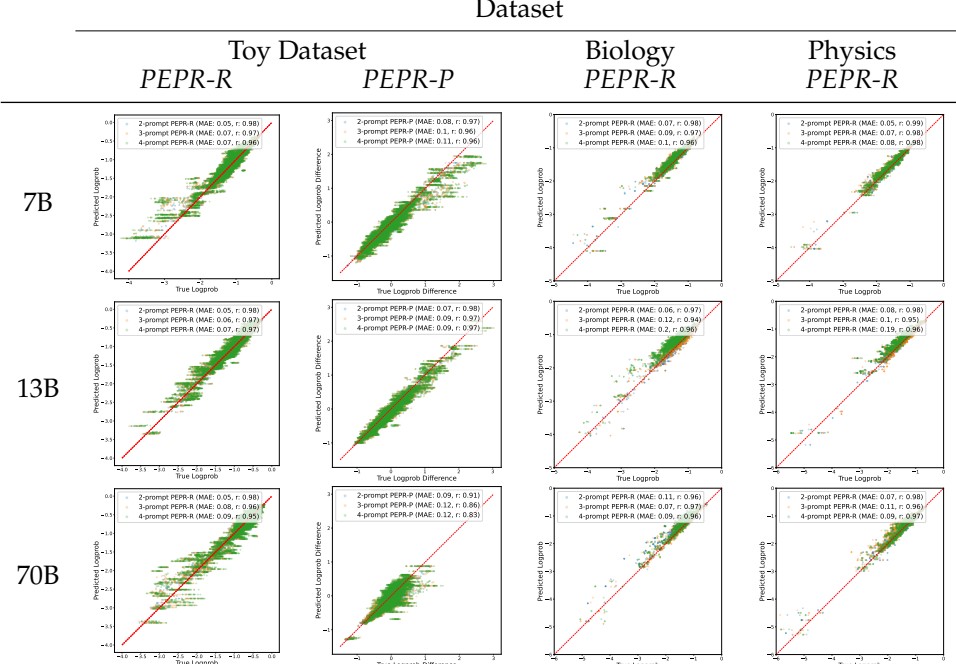

Figure 10: Predicted-versus-true value plots corresponding to prompt regression experiments on three datasets, our Toy Dataset, CAMEL Biology, and CAMEL Physics. Note that as the CAMEL datasets lack preference data, we only evaluate PEPR-R on those datasets.

## B.3 Main Experiment Prompt Selection Results

Table 3: Comprehensive accuracy results are reported in Table 3 with standard deviations pertaining to experiment settings. Models here are from the Llama-2-chat family. We include maximum and high percentile results from all relevant prompt combinations and baselines. Note that the high-performing prompts that PEPR suggests have small standard deviations, suggesting the stability of prompts selected as compared to the random method.

| | | Labeled Data Portion Method | | | | | | | | | Base | 0.75 | Max |
|---|---|---|---|---|---|---|---|---|---|---|---|---|---|
| | | 0.1 | | | 0.5 | | | 1 | | | | | |
| Dataset | Model | Rand | PEPR-R | PEPR-P | Rand | PEPR-R | PEPR-P | Rand | PEPR-R | PEPR-P | | | |
| Toy Dataset | 7B | 0.52 ± 0.04 | 0.41 ± 0.00 | **0.53 ± 0.00** | 0.52 ± 0.04 | 0.41 ± 0.00 | **0.53 ± 0.00** | 0.52 ± 0.04 | 0.41 ± 0.00 | **0.53 ± 0.00** | 0.19 | 0.37 | 0.55 |
| | 13B | 0.56 ± 0.05 | 0.56 ± 0.04 | **0.58 ± 0.01** | 0.56 ± 0.05 | 0.57 ± 0.00 | **0.58 ± 0.00** | 0.56 ± 0.05 | 0.57 ± 0.00 | **0.58 ± 0.00** | 0.16 | 0.40 | 0.61 |
| | 70B | 0.95 ± 0.06 | **0.97 ± 0.00** | **0.97 ± 0.00** | 0.95 ± 0.06 | **0.97 ± 0.00** | **0.97 ± 0.00** | 0.95 ± 0.06 | **0.97 ± 0.00** | **0.97 ± 0.00** | 0.16 | 0.67 | 0.98 |
| HateCheck (Slur) | 7B | **0.73 ± 0.02** | 0.68 ± 0.01 | 0.69 ± 0.01 | **0.73 ± 0.02** | 0.67 ± 0.00 | 0.68 ± 0.00 | **0.73 ± 0.02** | 0.67 ± 0.00 | 0.68 ± 0.00 | 0.65 | 0.69 | 0.76 |
| | 13B | **0.80 ± 0.02** | 0.75 ± 0.00 | **0.80 ± 0.02** | 0.80 ± 0.02 | 0.75 ± 0.00 | **0.81 ± 0.01** | 0.08 ± 0.02 | 0.75 ± 0.00 | **0.82 ± 0.00** | 0.80 | 0.75 | 0.83 |
| | 70B | **0.90 ± 0.03** | 0.90 ± 0.02 | 0.83 ± 0.02 | **0.90 ± 0.03** | 0.90 ± 0.00 | 0.83 ± 0.01 | **0.90 ± 0.03** | 0.90 ± 0.00 | 0.83 ± 0.00 | 0.71 | 0.84 | 0.95 |
| Biology | 7B | 0.57 ± 0.12 | **0.68 ± 0.00** | | 0.55 ± 0.12 | **0.68 ± 0.00** | | 0.55 ± 0.12 | **0.68 ± 0.00** | | 0.60 | 0.67 | 0.69 |
| | 13B | 0.57 ± 0.10 | **0.65 ± 0.07** | | 0.58 ± 0.10 | **0.68 ± 0.03** | | 0.58 ± 0.10 | **0.68 ± 0.00** | | 0.58 | 0.67 | 0.70 |
| | 70B | 0.47 ± 0.13 | **0.68 ± 0.00** | | 0.47 ± 0.14 | **0.68 ± 0.00** | | 0.47 ± 0.14 | **0.68 ± 0.00** | | 0.60 | 0.68 | 0.69 |
| Physics | 7B | 0.53 ± 0.09 | **0.65 ± 0.00** | | 0.54 ± 0.09 | **0.65 ± 0.00** | | 0.53 ± 0.10 | **0.65 ± 0.00** | | 0.53 | 0.63 | 0.65 |
| | 13B | 0.58 ± 0.08 | **0.66 ± 0.00** | | 0.57 ± 0.08 | **0.66 ± 0.00** | | 0.57 ± 0.08 | **0.66 ± 0.00** | | 0.54 | 0.65 | 0.66 |
| | 70B | 0.51 ± 0.12 | **0.66 ± 0.00** | | 0.50 ± 0.12 | **0.66 ± 0.00** | | 0.50 ± 0.12 | **0.66 ± 0.00** | | 0.61 | 0.63 | 0.66 |
| NI 020 | 7B | 0.69 ± 0.10 | 0.69 ± 0.10 | **0.70 ± 0.10** | **0.74 ± 0.01** | **0.74 ± 0.01** | **0.74 ± 0.01** | 0.74 ± 0.01 | **0.74 ± 0.00** | **0.74 ± 0.00** | 0.57 | 0.74 | 0.76 |
| | 13B | 0.71 ± 0.07 | 0.72 ± 0.05 | **0.74 ± 0.07** | 0.75 ± 0.02 | 0.76 ± 0.01 | **0.78 ± 0.00** | 0.76 ± 0.01 | 0.77 ± 0.00 | **0.78 ± 0.00** | 0.64 | 0.74 | 0.78 |
| | 70B | 0.71 ± 0.08 | 0.71 ± 0.08 | **0.74 ± 0.03** | 0.75 ± 0.03 | 0.74 ± 0.01 | **0.75 ± 0.00** | **0.76 ± 0.01** | 0.74 ± 0.00 | 0.75 ± 0.00 | 0.52 | 0.74 | 0.79 |
| NI 195 | 7B | **0.69 ± 0.10** | 0.60 ± 0.10 | 0.55 ± 0.05 | **0.74 ± 0.08** | 0.67 ± 0.02 | 0.56 ± 0.01 | **0.74 ± 0.08** | 0.67 ± 0.00 | 0.56 ± 0.00 | 0.56 | 0.57 | 0.85 |
| | 13B | **0.62 ± 0.08** | 0.59 ± 0.03 | 0.60 ± 0.05 | **0.64 ± 0.09** | 0.62 ± 0.01 | 0.62 ± 0.05 | 0.65 ± 0.08 | 0.62 ± 0.00 | **0.66 ± 0.00** | 0.56 | 0.56 | 0.83 |
| | 70B | 0.74 ± 0.07 | 0.75 ± 0.05 | **0.78 ± 0.07** | 0.79 ± 0.04 | 0.80 ± 0.03 | **0.81 ± 0.01** | 0.79 ± 0.03 | **0.81 ± 0.00** | **0.81 ± 0.00** | 0.56 | 0.71 | 0.84 |
| NI 199 | 7B | 0.74 ± 0.10 | **0.76 ± 0.06** | 0.74 ± 0.10 | **0.77 ± 0.00** | **0.77 ± 0.00** | **0.77 ± 0.00** | **0.77 ± 0.00** | **0.77 ± 0.00** | **0.77 ± 0.00** | 0.70 | 0.77 | 0.77 |
| | 13B | 0.74 ± 0.10 | **0.77 ± 0.00** | 0.77 ± 0.01 | 0.77 ± 0.01 | **0.77 ± 0.00** | **0.77 ± 0.00** | **0.77 ± 0.00** | **0.77 ± 0.00** | **0.77 ± 0.00** | 0.39 | 0.77 | 0.77 |
| | 70B | 0.74 ± 0.09 | 0.74 ± 0.07 | **0.74 ± 0.06** | **0.77 ± 0.06** | 0.76 ± 0.00 | 0.76 ± 0.00 | **0.77 ± 0.00** | 0.76 ± 0.00 | 0.76 ± 0.00 | 0.59 | 0.77 | 0.78 |

| | | Labeled Data Portion Method | | | | | | Base | 0.75 | Max |
|---|---|---|---|---|---|---|---|---|---|---|
| | | 0.05 | | | 0.25 | | | | | |
| Dataset | Model | Rand | PEPR-R | PEPR-P | Rand | PEPR-R | PEPR-P | | | |
| Toy Dataset | 7B | 0.52 ± 0.04 | 0.41 ± 0.01 | **0.53 ± 0.00** | 0.52 ± 0.04 | 0.41 ± 0.00 | **0.53 ± 0.00** | 0.19 | 0.37 | 0.55 |
| | 13B | 0.56 ± 0.05 | 0.56 ± 0.06 | **0.58 ± 0.01** | 0.56 ± 0.05 | 0.57 ± 0.00 | **0.58 ± 0.00** | 0.16 | 0.40 | 0.61 |
| | 70B | 0.95 ± 0.06 | **0.97 ± 0.00** | **0.97 ± 0.00** | 0.95 ± 0.06 | **0.97 ± 0.00** | **0.97 ± 0.00** | 0.16 | 0.67 | 0.98 |
| HateCheck (Slur) | 7B | **0.73 ± 0.02** | 0.68 ± 0.01 | 0.69 ± 0.01 | **0.73 ± 0.02** | 0.67 ± 0.00 | 0.68 ± 0.01 | 0.65 | 0.69 | 0.76 |
| | 13B | **0.80 ± 0.02** | 0.74 ± 0.01 | **0.80 ± 0.03** | 0.80 ± 0.02 | 0.75 ± 0.00 | **0.81 ± 0.02** | 0.80 | 0.75 | 0.83 |
| | 70B | **0.90 ± 0.03** | 0.89 ± 0.02 | 0.83 ± 0.03 | **0.90 ± 0.03** | 0.90 ± 0.00 | 0.83 ± 0.02 | 0.71 | 0.84 | 0.95 |
| Biology | 7B | 0.55 ± 0.12 | **0.68 ± 0.00** | | 0.55 ± 0.12 | **0.68 ± 0.00** | | 0.60 | 0.67 | 0.69 |
| | 13B | 0.58 ± 0.10 | **0.68 ± 0.03** | | 0.58 ± 0.10 | **0.68 ± 0.00** | | 0.58 | 0.67 | 0.70 |
| | 70B | 0.47 ± 0.14 | **0.68 ± 0.00** | | 0.47 ± 0.14 | **0.68 ± 0.00** | | 0.60 | 0.68 | 0.69 |
| Physics | 7B | 0.54 ± 0.09 | **0.65 ± 0.00** | | 0.53 ± 0.10 | **0.65 ± 0.00** | | 0.53 | 0.63 | 0.65 |
| | 13B | 0.57 ± 0.08 | **0.66 ± 0.00** | | 0.57 ± 0.08 | **0.66 ± 0.00** | | 0.54 | 0.65 | 0.66 |
| | 70B | 0.50 ± 0.12 | **0.66 ± 0.00** | | 0.50 ± 0.12 | **0.66 ± 0.00** | | 0.61 | 0.63 | 0.66 |
| NI 020 | 7B | **0.74 ± 0.01** | **0.74 ± 0.01** | **0.74 ± 0.01** | 0.74 ± 0.01 | **0.74 ± 0.00** | **0.74 ± 0.00** | 0.57 | 0.74 | 0.76 |
| | 13B | 0.75 ± 0.02 | 0.76 ± 0.01 | **0.78 ± 0.00** | 0.76 ± 0.01 | 0.77 ± 0.00 | **0.78 ± 0.00** | 0.64 | 0.74 | 0.78 |
| | 70B | 0.75 ± 0.03 | 0.74 ± 0.01 | **0.75 ± 0.00** | **0.76 ± 0.01** | 0.74 ± 0.00 | 0.75 ± 0.00 | 0.52 | 0.74 | 0.79 |
| NI 195 | 7B | **0.74 ± 0.08** | 0.67 ± 0.02 | 0.56 ± 0.01 | **0.74 ± 0.08** | 0.67 ± 0.00 | 0.56 ± 0.01 | 0.56 | 0.57 | 0.85 |
| | 13B | **0.64 ± 0.09** | 0.62 ± 0.01 | 0.62 ± 0.05 | 0.65 ± 0.08 | 0.62 ± 0.00 | **0.66 ± 0.00** | 0.56 | 0.56 | 0.83 |
| | 70B | 0.79 ± 0.04 | 0.80 ± 0.03 | **0.81 ± 0.01** | 0.79 ± 0.03 | **0.81 ± 0.00** | **0.81 ± 0.00** | 0.56 | 0.71 | 0.84 |
| NI 199 | 7B | **0.77 ± 0.00** | **0.77 ± 0.00** | **0.77 ± 0.00** | **0.77 ± 0.00** | **0.77 ± 0.00** | **0.77 ± 0.00** | 0.70 | 0.77 | 0.77 |
| | 13B | 0.77 ± 0.01 | **0.77 ± 0.00** | **0.77 ± 0.00** | **0.77 ± 0.00** | **0.77 ± 0.00** | **0.77 ± 0.00** | 0.39 | 0.77 | 0.77 |
| | 70B | **0.77 ± 0.06** | 0.76 ± 0.00 | 0.76 ± 0.00 | **0.77 ± 0.00** | 0.76 ± 0.00 | 0.76 ± 0.00 | 0.59 | 0.77 | 0.78 |

Table 4: Results on other 10 data classes from the HateCheck dataset. As with results in Table 3, standard deviations for prompts selected by PEPR are small, suggesting prompt stability. See (Röttger et al., 2020) for more details about each class.

| Data Class | Model | Labeled Data Portion / Method | | | | | | | | | Base | 0.75 | Max |
|---|---|---|---|---|---|---|---|---|---|---|---|---|---|
| | | 0.1 | | | 0.5 | | | 1 | | | | | |
| | | Rand | PEPR-R | PEPR-P | Rand | PEPR-R | PEPR-P | Rand | PEPR-R | PEPR-P | | | |
| Derogation | 7B | 0.98 ± 0.01 | 0.91 ± 0.00 | **0.99 ± 0.00** | 0.98 ± 0.01 | 0.91 ± 0.00 | **0.99 ± 0.00** | 0.98 ± 0.01 | 0.91 ± 0.00 | **0.99 ± 0.00** | 0.97 | 0.95 | 0.99 |
| | 13B | 0.94 ± 0.03 | 0.79 ± 0.00 | **0.96 ± 0.00** | 0.94 ± 0.03 | 0.79 ± 0.00 | **0.96 ± 0.00** | 0.94 ± 0.03 | 0.79 ± 0.00 | **0.96 ± 0.00** | 0.87 | 0.85 | 0.96 |
| | 70B | **1.00 ± 0.00** | **1.00 ± 0.00** | **1.00 ± 0.00** | **1.00 ± 0.00** | **1.00 ± 0.00** | **1.00 ± 0.00** | **1.00 ± 0.00** | **1.00 ± 0.00** | **1.00 ± 0.00** | 1.00 | 1.00 | 1.00 |
| Threatening language | 7B | **1.00 ± 0.00** | **1.00 ± 0.00** | **1.00 ± 0.00** | **1.00 ± 0.00** | **1.00 ± 0.00** | **1.00 ± 0.00** | **1.00 ± 0.00** | **1.00 ± 0.00** | **1.00 ± 0.00** | 1.00 | 1.00 | 1.00 |
| | 13B | 0.99 ± 0.00 | 0.95 ± 0.00 | **1.00 ± 0.00** | 0.99 ± 0.00 | 0.95 ± 0.00 | **1.00 ± 0.00** | 0.99 ± 0.00 | 0.95 ± 0.00 | **1.00 ± 0.00** | 0.99 | 0.98 | 1.00 |
| | 70B | **1.00 ± 0.00** | **1.00 ± 0.00** | **1.00 ± 0.00** | **1.00 ± 0.00** | **1.00 ± 0.00** | **1.00 ± 0.00** | **1.00 ± 0.00** | **1.00 ± 0.00** | **1.00 ± 0.00** | 1.00 | 1.00 | 1.00 |
| Profanity usage | 7B | **0.97 ± 0.01** | 0.95 ± 0.00 | 0.96 ± 0.02 | **0.97 ± 0.01** | 0.94 ± 0.00 | 0.96 ± 0.00 | **0.97 ± 0.01** | 0.94 ± 0.00 | 0.96 ± 0.00 | 0.95 | 0.95 | 0.98 |
| | 13B | 0.97 ± 0.01 | 0.96 ± 0.00 | **0.98 ± 0.00** | 0.97 ± 0.01 | 0.96 ± 0.00 | **0.98 ± 0.00** | 0.97 ± 0.01 | 0.96 ± 0.00 | **0.98 ± 0.00** | 0.95 | 0.95 | 0.98 |
| | 70B | **1.00 ± 0.00** | **1.00 ± 0.00** | 0.96 ± 0.03 | **1.00 ± 0.00** | **1.00 ± 0.00** | 0.95 ± 0.02 | **1.00 ± 0.00** | **1.00 ± 0.00** | 0.93 ± 0.00 | 0.90 | 0.99 | 1.00 |
| Pronoun reference | 7B | 0.99 ± 0.01 | 0.95 ± 0.00 | **1.00 ± 0.00** | 0.99 ± 0.01 | 0.95 ± 0.00 | **1.00 ± 0.00** | 0.99 ± 0.01 | 0.95 ± 0.00 | **1.00 ± 0.00** | 0.97 | 0.96 | 1.00 |
| | 13B | 0.98 ± 0.01 | 0.91 ± 0.00 | **0.99 ± 0.00** | 0.98 ± 0.01 | 0.91 ± 0.00 | **0.99 ± 0.00** | 0.98 ± 0.01 | 0.91 ± 0.00 | **0.99 ± 0.00** | 0.93 | 0.94 | 1.00 |
| | 70B | **1.00 ± 0.00** | **1.00 ± 0.00** | **1.00 ± 0.00** | **1.00 ± 0.00** | **1.00 ± 0.00** | **1.00 ± 0.00** | **1.00 ± 0.00** | **1.00 ± 0.00** | **1.00 ± 0.00** | 1.00 | 1.00 | 1.00 |
| Negation | 7B | **0.96 ± 0.01** | 0.95 ± 0.01 | **0.96 ± 0.00** | **0.96 ± 0.01** | 0.95 ± 0.00 | **0.96 ± 0.00** | **0.96 ± 0.01** | 0.95 ± 0.00 | **0.96 ± 0.00** | 0.87 | 0.93 | 0.98 |
| | 13B | **0.96 ± 0.00** | 0.92 ± 0.00 | 0.95 ± 0.00 | **0.96 ± 0.00** | 0.92 ± 0.00 | 0.95 ± 0.00 | **0.96 ± 0.00** | 0.92 ± 0.00 | 0.95 ± 0.00 | 0.96 | 0.95 | 0.97 |
| | 70B | **0.99 ± 0.00** | 0.98 ± 0.00 | 0.97 ± 0.01 | **0.99 ± 0.00** | 0.98 ± 0.00 | 0.97 ± 0.00 | **0.99 ± 0.00** | 0.98 ± 0.00 | 0.97 ± 0.00 | 0.89 | 0.98 | 1.00 |
| Phrasing | 7B | 0.99 ± 0.01 | 0.86 ± 0.00 | **1.00 ± 0.00** | 0.99 ± 0.01 | 0.86 ± 0.00 | **1.00 ± 0.00** | 0.99 ± 0.01 | 0.86 ± 0.00 | **1.00 ± 0.00** | 0.98 | 0.97 | 1.00 |
| | 13B | 0.97 ± 0.02 | 0.81 ± 0.00 | **0.99 ± 0.00** | 0.97 ± 0.02 | 0.81 ± 0.00 | **0.99 ± 0.00** | 0.97 ± 0.02 | 0.81 ± 0.00 | **0.99 ± 0.00** | 0.95 | 0.90 | 1.00 |
| | 70B | **1.00 ± 0.00** | **1.00 ± 0.00** | **1.00 ± 0.00** | **1.00 ± 0.00** | **1.00 ± 0.00** | **1.00 ± 0.00** | **1.00 ± 0.00** | **1.00 ± 0.00** | **1.00 ± 0.00** | 1.00 | 1.00 | 1.00 |
| Non-hate group identifiers | 7B | **1.00 ± 0.00** | 0.99 ± 0.01 | **1.00 ± 0.00** | **1.00 ± 0.00** | **1.00 ± 0.00** | **1.00 ± 0.00** | **1.00 ± 0.00** | **1.00 ± 0.00** | **1.00 ± 0.00** | 0.98 | 0.97 | 1.00 |
| | 13B | **1.00 ± 0.00** | **1.00 ± 0.00** | **1.00 ± 0.00** | **1.00 ± 0.00** | **1.00 ± 0.00** | **1.00 ± 0.00** | **1.00 ± 0.00** | **1.00 ± 0.00** | **1.00 ± 0.00** | 0.98 | 1.00 | 1.00 |
| | 70B | **0.99 ± 0.00** | 0.98 ± 0.00 | 0.98 ± 0.00 | **0.99 ± 0.00** | 0.98 ± 0.00 | 0.98 ± 0.00 | **0.99 ± 0.00** | 0.98 ± 0.00 | 0.98 ± 0.00 | 1.00 | 1.00 | 1.00 |
| Counter speech | 7B | **0.94 ± 0.04** | 0.79 ± 0.04 | 0.80 ± 0.11 | **0.94 ± 0.04** | 0.81 ± 0.00 | 0.71 ± 0.04 | **0.94 ± 0.04** | 0.81 ± 0.00 | 0.70 ± 0.00 | 0.15 | 0.79 | 0.98 |
| | 13B | **0.98 ± 0.03** | 0.94 ± 0.00 | 0.91 ± 0.02 | **0.98 ± 0.03** | 0.94 ± 0.00 | 0.91 ± 0.00 | **0.98 ± 0.03** | 0.94 ± 0.00 | 0.91 ± 0.00 | 0.34 | 0.86 | 1.00 |
| | 70B | 0.58 ± 0.08 | 0.55 ± 0.19 | **0.62 ± 0.00** | 0.58 ± 0.08 | 0.28 ± 0.00 | **0.62 ± 0.00** | 0.58 ± 0.08 | 0.28 ± 0.00 | **0.62 ± 0.00** | 0.04 | 0.36 | 0.71 |
| Abuse against non-prot. targets | 7B | **0.99 ± 0.02** | 0.90 ± 0.00 | 0.98 ± 0.03 | **0.99 ± 0.02** | 0.91 ± 0.00 | 0.99 ± 0.00 | **0.99 ± 0.02** | 0.91 ± 0.00 | 0.99 ± 0.00 | 0.61 | 0.90 | 1.00 |
| | 13B | **0.99 ± 0.02** | 0.93 ± 0.01 | 0.88 ± 0.01 | **0.99 ± 0.02** | 0.93 ± 0.00 | 0.92 ± 0.03 | **0.99 ± 0.02** | 0.93 ± 0.00 | 0.95 ± 0.00 | 0.74 | 0.93 | 1.00 |
| | 70B | **0.75 ± 0.06** | 0.66 ± 0.00 | 0.68 ± 0.04 | **0.75 ± 0.06** | 0.66 ± 0.00 | 0.67 ± 0.03 | **0.75 ± 0.06** | 0.66 ± 0.00 | 0.65 ± 0.00 | 0.49 | 0.62 | 0.87 |
| Spelling variations | 7B | 0.93 ± 0.01 | 0.83 ± 0.00 | **0.93 ± 0.00** | 0.93 ± 0.01 | 0.83 ± 0.00 | **0.93 ± 0.00** | 0.93 ± 0.01 | 0.83 ± 0.00 | **0.93 ± 0.00** | 0.91 | 0.90 | 0.96 |
| | 13B | 0.89 ± 0.03 | 0.76 ± 0.00 | **0.93 ± 0.00** | 0.89 ± 0.03 | 0.76 ± 0.00 | **0.93 ± 0.00** | 0.89 ± 0.03 | 0.76 ± 0.00 | **0.93 ± 0.00** | 0.84 | 0.81 | 0.94 |
| | 70B | **1.00 ± 0.00** | **1.00 ± 0.00** | **1.00 ± 0.00** | **1.00 ± 0.00** | **1.00 ± 0.00** | **1.00 ± 0.00** | **1.00 ± 0.00** | **1.00 ± 0.00** | **1.00 ± 0.00** | 1.00 | 1.00 | 1.00 |

| Data Class | Model | Labeled Data Portion / Method | | | | | | Base | 0.75 | Max |
|---|---|---|---|---|---|---|---|---|---|---|
| | | 0.05 | | | 0.25 | | | | | |
| | | Rand | PEPR-R | PEPR-P | Rand | PEPR-R | PEPR-P | | | |
| Derogation | 7B | **0.98 ± 0.01** | 0.91 ± 0.00 | **0.98 ± 0.00** | 0.98 ± 0.01 | 0.91 ± 0.00 | **0.99 ± 0.00** | 0.97 | 0.95 | 0.99 |
| | 13B | 0.94 ± 0.03 | 0.79 ± 0.00 | **0.96 ± 0.00** | 0.94 ± 0.03 | 0.79 ± 0.00 | **0.96 ± 0.00** | 0.87 | 0.85 | 0.96 |
| | 70B | **1.00 ± 0.00** | **1.00 ± 0.00** | **1.00 ± 0.00** | **1.00 ± 0.00** | **1.00 ± 0.00** | **1.00 ± 0.00** | 1.00 | 1.00 | 1.00 |
| Threatening language | 7B | **1.00 ± 0.00** | **1.00 ± 0.00** | **1.00 ± 0.00** | **1.00 ± 0.00** | **1.00 ± 0.00** | **1.00 ± 0.00** | 1.00 | 1.00 | 1.00 |
| | 13B | 0.99 ± 0.00 | 0.95 ± 0.00 | **1.00 ± 0.00** | 0.99 ± 0.00 | 0.95 ± 0.00 | **1.00 ± 0.00** | 0.99 | 0.98 | 1.00 |
| | 70B | **1.00 ± 0.00** | **1.00 ± 0.00** | **1.00 ± 0.00** | **1.00 ± 0.00** | **1.00 ± 0.00** | **1.00 ± 0.00** | 1.00 | 1.00 | 1.00 |
| Profanity usage | 7B | **0.97 ± 0.01** | 0.95 ± 0.00 | 0.96 ± 0.02 | **0.97 ± 0.01** | 0.95 ± 0.00 | 0.96 ± 0.01 | 0.95 | 0.95 | 0.98 |
| | 13B | 0.97 ± 0.01 | 0.96 ± 0.00 | **0.98 ± 0.00** | 0.97 ± 0.01 | 0.96 ± 0.00 | **0.98 ± 0.00** | 0.95 | 0.95 | 0.98 |
| | 70B | **1.00 ± 0.00** | **1.00 ± 0.00** | 0.96 ± 0.03 | **1.00 ± 0.00** | **1.00 ± 0.00** | 0.96 ± 0.03 | 0.90 | 0.99 | 1.00 |
| Pronoun reference | 7B | 0.99 ± 0.01 | 0.95 ± 0.00 | **1.00 ± 0.00** | 0.99 ± 0.01 | 0.95 ± 0.00 | **1.00 ± 0.00** | 0.97 | 0.96 | 1.00 |
| | 13B | 0.98 ± 0.01 | 0.91 ± 0.00 | **0.99 ± 0.00** | 0.98 ± 0.01 | 0.91 ± 0.00 | **0.99 ± 0.00** | 0.93 | 0.94 | 1.00 |
| | 70B | **1.00 ± 0.00** | **1.00 ± 0.00** | **1.00 ± 0.00** | **1.00 ± 0.00** | **1.00 ± 0.00** | **1.00 ± 0.00** | 1.00 | 1.00 | 1.00 |
| Negation | 7B | **0.96 ± 0.01** | 0.95 ± 0.01 | **0.96 ± 0.00** | **0.96 ± 0.01** | 0.95 ± 0.00 | **0.96 ± 0.00** | 0.87 | 0.93 | 0.98 |
| | 13B | **0.96 ± 0.00** | 0.92 ± 0.00 | 0.95 ± 0.01 | **0.96 ± 0.00** | 0.92 ± 0.00 | 0.95 ± 0.00 | 0.96 | 0.95 | 0.97 |
| | 70B | **0.99 ± 0.00** | 0.98 ± 0.01 | 0.98 ± 0.01 | **0.99 ± 0.00** | 0.98 ± 0.00 | 0.97 ± 0.00 | 0.89 | 0.98 | 1.00 |
| Phrasing | 7B | 0.99 ± 0.01 | 0.86 ± 0.00 | **1.00 ± 0.00** | 0.99 ± 0.01 | 0.86 ± 0.00 | **1.00 ± 0.00** | 0.98 | 0.97 | 1.00 |
| | 13B | 0.97 ± 0.02 | 0.81 ± 0.00 | **0.99 ± 0.00** | 0.97 ± 0.02 | 0.81 ± 0.00 | **0.99 ± 0.00** | 0.95 | 0.90 | 1.00 |
| | 70B | **1.00 ± 0.00** | **1.00 ± 0.00** | **1.00 ± 0.00** | **1.00 ± 0.00** | **1.00 ± 0.00** | **1.00 ± 0.00** | 1.00 | 1.00 | 1.00 |
| Non-hate group identifiers | 7B | **1.00 ± 0.00** | 0.99 ± 0.02 | **1.00 ± 0.00** | **1.00 ± 0.00** | 1.00 ± 0.01 | **1.00 ± 0.00** | 0.98 | 0.97 | 1.00 |
| | 13B | **1.00 ± 0.00** | **1.00 ± 0.00** | **1.00 ± 0.00** | **1.00 ± 0.00** | **1.00 ± 0.00** | **1.00 ± 0.00** | 0.98 | 1.00 | 1.00 |
| | 70B | **0.99 ± 0.00** | 0.98 ± 0.00 | 0.98 ± 0.00 | **0.99 ± 0.00** | 0.98 ± 0.00 | 0.98 ± 0.00 | 1.00 | 1.00 | 1.00 |
| Counter speech | 7B | **0.94 ± 0.04** | 0.78 ± 0.05 | 0.82 ± 0.11 | **0.94 ± 0.04** | 0.81 ± 0.00 | 0.74 ± 0.08 | 0.15 | 0.79 | 0.98 |
| | 13B | **0.98 ± 0.03** | 0.94 ± 0.00 | 0.90 ± 0.03 | **0.98 ± 0.03** | 0.94 ± 0.00 | 0.91 ± 0.01 | 0.34 | 0.86 | 1.00 |
| | 70B | 0.58 ± 0.08 | **0.68 ± 0.03** | 0.62 ± 0.00 | 0.58 ± 0.08 | 0.28 ± 0.00 | **0.62 ± 0.00** | 0.04 | 0.36 | 0.71 |
| Abuse against non-prot. targets | 7B | **0.99 ± 0.02** | 0.90 ± 0.00 | 0.97 ± 0.04 | **0.99 ± 0.02** | 0.90 ± 0.00 | **0.99 ± 0.01** | 0.61 | 0.90 | 1.00 |
| | 13B | **0.99 ± 0.02** | 0.94 ± 0.01 | 0.88 ± 0.02 | **0.99 ± 0.02** | 0.93 ± 0.00 | 0.88 ± 0.00 | 0.74 | 0.93 | 1.00 |
| | 70B | **0.75 ± 0.06** | 0.66 ± 0.00 | 0.68 ± 0.04 | **0.75 ± 0.06** | 0.66 ± 0.00 | 0.67 ± 0.03 | 0.49 | 0.62 | 0.87 |
| Spelling variations | 7B | **0.93 ± 0.01** | 0.84 ± 0.01 | **0.93 ± 0.00** | **0.93 ± 0.01** | 0.83 ± 0.00 | **0.93 ± 0.00** | 0.91 | 0.90 | 0.96 |
| | 13B | 0.89 ± 0.03 | 0.76 ± 0.00 | **0.93 ± 0.00** | 0.89 ± 0.03 | 0.76 ± 0.00 | **0.93 ± 0.00** | 0.84 | 0.81 | 0.94 |
| | 70B | **1.00 ± 0.00** | **1.00 ± 0.00** | **1.00 ± 0.00** | **1.00 ± 0.00** | **1.00 ± 0.00** | **1.00 ± 0.00** | 1.00 | 1.00 | 1.00 |

Table 5: Subset of selection experiment results with TRIPLE-SH baseline. Note that TRIPLE-SH results could not be computed for some smaller values of dataset portions due to too few samples. Evidently, though TRIPLE-SH outperforms both versions of PEPR for some models and datasets, PEPR can be used with small amounts of data (i.e., limited evaluation budget) and is arguably more interpretable.

| | | Labeled Data Portion | | | | | | | | | | Method | | | |
| | | 0.05 | | | 0.25 | | | 0.5 | | | 1 | | | | | |
| Dataset | Model | TR-SH | PEPR-R | PEPR-P | TR-SH | PEPR-R | PEPR-P | TR-SH | PEPR-R | PEPR-P | TR-SH | PEPR-R | PEPR-P | Base | 0.75 | Max |
|---|---|---|---|---|---|---|---|---|---|---|---|---|---|---|---|---|
| Toy Dataset | 7B | 0.39 | 0.41 | **0.53** | 0.51 | 0.41 | **0.53** | 0.52 | 0.41 | **0.53** | **0.54** | 0.41 | 0.53 | 0.19 | 0.37 | 0.55 |
| | 13B | 0.48 | 0.56 | **0.58** | 0.56 | 0.57 | **0.58** | 0.57 | 0.57 | **0.58** | **0.59** | 0.57 | 0.58 | 0.16 | 0.40 | 0.61 |
| | 70B | 0.82 | 0.97 | 0.97 | 0.84 | **0.97** | 0.97 | 0.86 | **0.97** | 0.97 | 0.88 | **0.97** | 0.97 | 0.16 | 0.67 | 0.98 |
| HateCheck (Slur) | 7B | N/A | 0.68 | 0.69 | **0.69** | 0.67 | 0.68 | **0.72** | 0.67 | 0.68 | **0.73** | 0.67 | 0.68 | 0.65 | 0.69 | 0.76 |
| | 13B | N/A | 0.74 | **0.80** | 0.80 | 0.75 | **0.81** | **0.81** | 0.75 | **0.81** | 0.81 | 0.75 | **0.82** | 0.80 | 0.75 | 0.83 |
| | 70B | N/A | **0.90** | 0.83 | **0.93** | 0.90 | 0.83 | **0.94** | 0.90 | 0.83 | **0.94** | 0.90 | 0.83 | 0.71 | 0.84 | 0.95 |
| Biology | 7B | N/A | **0.67** | | **0.68** | 0.68 | | **0.69** | 0.68 | | **0.68** | 0.68 | | 0.60 | 0.67 | 0.69 |
| | 13B | N/A | **0.64** | | **0.69** | 0.66 | | **0.70** | 0.68 | | **0.70** | 0.68 | | 0.58 | 0.67 | 0.70 |
| | 70B | N/A | **0.68** | | **0.69** | 0.68 | | **0.69** | 0.68 | | **0.69** | 0.68 | | 0.60 | 0.68 | 0.69 |
| Physics | 7B | N/A | **0.58** | | **0.65** | 0.58 | | **0.65** | 0.58 | | **0.65** | 0.58 | | 0.53 | 0.63 | 0.65 |
| | 13B | N/A | **0.52** | | **0.66** | 0.53 | | **0.66** | 0.53 | | **0.66** | 0.53 | | 0.54 | 0.65 | 0.66 |
| | 70B | N/A | **0.66** | | **0.66** | 0.66 | | **0.66** | **0.66** | | **0.66** | **0.66** | | 0.61 | 0.63 | 0.66 |
| NI 020 | 7B | N/A | **0.64** | 0.64 | **0.74** | 0.72 | 0.73 | **0.74** | 0.74 | 0.74 | **0.74** | 0.74 | 0.74 | 0.57 | 0.74 | 0.76 |
| | 13B | N/A | **0.71** | 0.71 | 0.75 | 0.75 | **0.77** | 0.74 | 0.76 | **0.78** | 0.76 | 0.77 | **0.78** | 0.64 | 0.74 | 0.79 |
| | 70B | N/A | 0.67 | **0.72** | **0.76** | 0.71 | 0.74 | **0.77** | 0.74 | 0.75 | **0.77** | 0.74 | 0.75 | 0.52 | 0.74 | 0.79 |
| NI 195 | 7B | N/A | **0.60** | 0.55 | **0.77** | 0.65 | 0.56 | **0.83** | 0.67 | 0.56 | **0.83** | 0.67 | 0.56 | 0.56 | 0.57 | 0.85 |
| | 13B | N/A | 0.58 | **0.59** | **0.77** | 0.61 | 0.61 | **0.79** | 0.62 | 0.62 | **0.79** | 0.62 | 0.66 | 0.56 | 0.56 | 0.83 |
| | 70B | N/A | **0.76** | 0.75 | **0.81** | 0.79 | 0.80 | **0.81** | 0.80 | **0.81** | 0.81 | **0.81** | **0.81** | 0.56 | 0.71 | 0.84 |
| NI 199 | 7B | N/A | **0.75** | 0.72 | **0.77** | **0.77** | **0.77** | **0.77** | **0.77** | **0.77** | **0.77** | **0.77** | **0.77** | 0.70 | 0.77 | 0.77 |
| | 13B | N/A | **0.76** | **0.76** | **0.77** | 0.76 | 0.76 | **0.77** | **0.77** | **0.77** | **0.77** | **0.77** | **0.77** | 0.39 | 0.77 | 0.77 |
| | 70B | N/A | 0.69 | **0.70** | 0.74 | **0.76** | **0.76** | **0.77** | 0.76 | 0.76 | **0.77** | 0.76 | 0.76 | 0.59 | 0.77 | 0.78 |

## B.4 Supplemental Experiments

**Mistral-7B-Instruct-v0.2** We test our method with an additional model, Mistral AI's Mistral 7B Instruct v0.2 (Jiang et al., 2023), on a subset of datasets used in our main experiments. More specifically, we do so on all datasets except our Toy Dataset and HateCheck.

Table 6: Mistral-7B-Instruct-v0.2 results reported in Table 6 with standard deviations pertaining to experiment settings. We include maximum and high percentile results from all relevant prompt combinations and baselines.

| | Labeled Data Portion | | | | | | | | | | | |
| | Method | | | | | | | | | | | |
| | 0.1 | | | 0.5 | | | 1 | | | | | |
| Dataset | Rand | PEPR-R | PEPR-P | Rand | PEPR-R | PEPR-P | Rand | PEPR-R | PEPR-P | Base | 0.75 | Max |
|---|---|---|---|---|---|---|---|---|---|---|---|---|
| Biology | 0.55 ± 0.10 | **0.68 ± 0.01** | | 0.53 ± 0.10 | **0.68 ± 0.00** | | 0.55 ± 0.10 | **0.68 ± 0.00** | | 0.56 | 0.68 | 0.70 |
| Physics | 0.54 ± 0.08 | **0.66 ± 0.00** | | 0.54 ± 0.08 | **0.65 ± 0.00** | | 0.54 ± 0.08 | **0.65 ± 0.00** | | 0.50 | 0.65 | 0.66 |
| NI 020 | 0.77 ± 0.03 | 0.78 ± 0.02 | **0.79 ± 0.01** | **0.79 ± 0.01** | **0.79 ± 0.01** | **0.79 ± 0.01** | **0.80 ± 0.00** | **0.80 ± 0.00** | **0.80 ± 0.00** | 0.76 | 0.78 | 0.82 |
| NI 195 | **0.76 ± 0.07** | 0.73 ± 0.03 | 0.71 ± 0.03 | **0.80 ± 0.03** | 0.76 ± 0.02 | 0.72 ± 0.03 | **0.80 ± 0.03** | 0.77 ± 0.00 | 0.73 ± 0.00 | 0.56 | 0.71 | 0.83 |
| NI 199 | 0.56 ± 0.12 | 0.40 ± 0.04 | **0.57 ± 0.14** | 0.61 ± 0.09 | 0.40 ± 0.02 | **0.69 ± 0.09** | 0.61 ± 0.08 | 0.39 ± 0.00 | **0.72 ± 0.00** | 0.36 | 0.44 | 0.73 |

| | Labeled Data Portion | | | | | | | | |
| | Method | | | | | | | | |
| | 0.05 | | | 0.25 | | | | | |
| Dataset | Rand | PEPR-R | PEPR-P | Rand | PEPR-R | PEPR-P | Base | 0.75 | Max |
|---|---|---|---|---|---|---|---|---|---|
| Biology | 0.54 ± 0.09 | **0.69 ± 0.01** | | 0.54 ± 0.10 | **0.68 ± 0.00** | | 0.56 | 0.68 | 0.70 |
| Physics | 0.54 ± 0.08 | **0.66 ± 0.00** | | 0.55 ± 0.08 | **0.65 ± 0.00** | | 0.50 | 0.65 | 0.66 |
| NI 020 | 0.77 ± 0.04 | **0.78 ± 0.02** | **0.78 ± 0.02** | 0.78 ± 0.02 | 0.79 ± 0.01 | **0.79 ± 0.00** | 0.76 | 0.78 | 0.82 |
| NI 195 | **0.73 ± 0.08** | 0.72 ± 0.03 | 0.70 ± 0.03 | **0.78 ± 0.04** | 0.75 ± 0.03 | 0.72 ± 0.03 | 0.56 | 0.71 | 0.83 |
| NI 199 | 0.52 ± 0.13 | 0.40 ± 0.04 | **0.57 ± 0.14** | 0.60 ± 0.10 | 0.40 ± 0.03 | **0.65 ± 0.13** | 0.36 | 0.44 | 0.73 |

