# OpenReview forum: "Prompt Exploration with Prompt Regression"
_colmweb.org/COLM/2024/Conference — COLM_

### Official Review · Reviewer_X964 · 2024-05-03

**Rating:** 6
**Confidence:** 3
**Ethics Flag:** 1

**Summary:**

This paper investigates how to select the optimal prompt from a prompt library given a specific model and a task. The authors propose a prompt regression approach, which utilizes task datasets to fit prompt weights, thereby achieving prompt selection. The authors design regression methods based on log-probability and preference for two types of task datasets, respectively. Overall, the methods proposed in the paper outperform baseline models, demonstrating their effectiveness.

**Questions To Authors:**

- The objective of the proposed methods is to learn prompt weights through regression conducted on specific datasets. Once these weights are obtained, they can be applied to each data point in the test set. My concern is whether this approach is task-specific. Do we need to know which task category the input belongs to in order to apply the corresponding prompt weights? If so, does this mean that such methods primarily apply to narrow-domain tasks, while being unsuitable for open-domain dialogue scenarios?

- During training, are multiple datasets mixed for training, or are they trained separately?

- Table 2 should be placed on page 8.

- From the results in Table 2, it seems that the effectiveness of PEPR may be related to the dataset. For some datasets, such as HateCheck and NI 199, even the "random" prompt achieves good results. Although the authors also provide some explanations for this phenomenon in the Discussion section, is it possible that certain tasks are insensitive to prompts?

- Previous studies have shown that large language models commonly suffer from overconfidence issues, wherein even incorrect or illusory generated results are assigned a high probability. Could this potentially affect methods based on log-probability?

- Please provide more experimental details for both the prompt regression and prompt selection to make it more clear and understandable.

- Please explain how the experimental results in Figure 2 validate the assumption that " the elimination or addition of an element from the prompt library does not affect interactions between the other prompts."

- Typo

  - Introduction section, " rigorously in the following sections," => "rigorously in the following sections." and "selection components of PEPR," => "selection components of PEPR."

**Reasons To Accept:**

- The proposed methods are reasonable, and the results demonstrate their effectiveness.

- The proposed method is derived from regression theory and is supported by mathematical definitions and formulations.

**Reasons To Reject:**

- The effectiveness of the methods may vary depending on the nature of the task.

- The proposed method didn't compare with other related works, comparing only with its baselines.

- In some cases, even random prompt combinations can outperform both versions of PEPR.

---

> ### Author Rebuttal · Authors · 2024-05-29
>
> We thank you for your evaluation of our work and overall positive sentiment. We will work to address the edits you have suggested provided our paper is accepted.
>
> > My concern is whether this approach is task-specific.
>
> PEPR allows prompt elements to depend on the input (footnote 1), thus the same prompt library can be used across different tasks of related nature, e.g., in Figure 5 we show such a prompt library used for both CAMEL Biology and Physics datasets. This brings in a certain degree of generality, but we note that prompt engineering is typically a task-specific problem.
>
> > Are multiple datasets mixed for training?
>
> We don’t mix HateCheck and CAMEL, but we do mix categories within them (like Biology and Physics for CAMEL) when fitting prompt regression (prompt selection is per category).
>
> > Overconfidence in LLMs
>
> We model relative differences in log-probability, so overconfidence might not be an issue if some prompt makes the LLM less overconfident.
>
> > Results in Figure 2 validate the assumption that "the elimination or addition of an element from the prompt library does not affect interactions between the other prompts."
>
> Figure 2 validates that a linear model (corresponding to independence of irrelevant alternatives) holds because of high correlation and low MAE. If interactions were to have a major impact, a linear model would not do well.
>
> > The proposed method didn't compare with other related works, comparing only with its baselines.
>
> We report 75th percentile and maximum performance as a proxy for budget-constrained searches seen in related work, but we can also report results of the method proposed by Shi et al. (2024) in a revision. Note that prior works are typically limited to around 100 prompt variants and they need to evaluate each at least a few times. Our method models the compositional structure of prompts and we applied it to over 1000 prompt variations (a prompt library with 10 elements) without evaluating each variant even once.
>
> > In some cases, even random prompt combinations can outperform both versions of PEPR. Is it possible that certain tasks are insensitive to prompts?
>
> Random prompt combinations is a fairly strong baseline if the prompt library is good. In such cases both “random” and PEPR always outperform the base model and 75\% percentile in most cases, suggesting that prompts always have an effect. Our method is able to filter out unhelpful prompt elements, while prompt regression also facilitates interpretability.

---

> > ### Comment · Reviewer_X964 · 2024-06-04
> >
> > hi, thank you for your response. I'd like to see this submission accepted, though I stick to my rating score.

---

> > > ### Author Response · Authors · 2024-06-05
> > >
> > > Thank you for your feedback and for considering the rebuttal.

---

### Official Review · Reviewer_jCnP · 2024-05-09

**Rating:** 6
**Confidence:** 4
**Ethics Flag:** 1

**Summary:**

The paper tackles a specific prompt engineering problem for LLMs, aiming to optimize prompt combinations from a library of individual elements. The goal is to predict the effects of these combinations on LLM outputs and create an optimal prompt for specific tasks. The proposed solution, Prompt Exploration with Prompt Regression (PEPR), consists of three steps: building a prompt library, deriving weights for each element through prompt regression, and selecting prompt elements via these weights. This process incorporates generated texts or human preferences to align prompts with the desired outcomes. PEPR's effectiveness is validated using several open-source LLMs across different datasets and tasks.

**Questions To Authors:**

1. In the prompt selection experiments illustrated in Figure 2, regression performance deteriorates as the number of prompt elements increases, indicated by larger MAE and smaller r-values. What could be the underlying reasons for this trend? Does this imply that an increase in the number of prompt elements leads to more coupling between them, thereby reducing their independence? Moreover, does the extent of this coupling depend on the quality of the prompt library?

2. What exactly is the "minimally-prompted model" mentioned in the second paragraph of section 4.2? Why was this baseline chosen instead of more advanced methods summarized in the related work section? What are the implications of using such a baseline for the validity of the experimental results?

**Reasons To Accept:**

1. The paper is commendable for its comprehensive review and summarization of up-to-date related works, effectively highlighting novelty of the paper.

2. The quality of writing in this paper is excellent. It is exceptionally clear and well-structured, making it easy for readers to understand and follow the arguments and methodologies presented.

3. The authors have included a detailed appendix which enriches the paper further. This appendix serves as a valuable resource for readers seeking a deeper understanding of the paper's contributions and the intricacies of the methodologies used.

**Reasons To Reject:**

1. The central concern of the paper is the prompt library search problem within prompt engineering, which seems under-motivated. The quality of the final prompt is inherently limited by the quality of the constructed prompt library, particularly in terms of coverage and richness. Moreover, PEPR's approach to determining the weight of each prompt element through parameter optimization on specific data significantly constrains the generalizability of the resultant prompts. Additionally, the paper does not adequately discuss the scale of data used for parameter optimization, leaving PEPR's practical applicability unclear.

2. The evaluation and rationalization of PEPR's effectiveness are incomplete. The experiments on prompt regression merely test the assumption of the independence of irrelevant alternatives, while those on prompt selection only assess PEPR's impact on task performance under various settings. Crucially, there is a lack of direct comparison between PEPR and the advanced prompt engineering methods described in the related work section. Such comparisons are vital to directly demonstrate PEPR's superiority and the relevance of analyzing prompt element combinations for producing the optimal overall prompt as outlined in the initial sections of the paper.

3. More importantly, current experimental results (referenced in Table 2) show that PEPR barely outperforms or does not surpass the random baseline. As expressed in the last paragraph of section 4.2, even with performance metrics variations across different datasets, these outcomes further challenge the significance and motivation behind the paper's focus on the prompt library search problem.

---

> ### Author Rebuttal · Authors · 2024-05-29
>
> Thank you for your review. We appreciate your positive points and the chance to respond to your questions and concerns.
>
> > Motivation for prompt library search problem
>
> System prompt selection is often a process of adding phrases to correct undesired behavior. Libraries of such phrases can grow big, thus requiring a method to identify an optimal combination algorithmically.
>
> > Generalizability of the PEPR prompts
>
> PEPR allows prompt elements to depend on the input (footnote 1), thus the same prompt library can be used across different tasks of related nature, e.g., in Figure 5 we show such a prompt library used for both CAMEL Biology and Physics datasets. In comparison to prior prompt-engineering approaches which are often task-specific, this feature increases the generalizability of PEPR.
>
> > Scale of data used for parameter optimization
>
> In Sections 3.2 and 4.2 we specify that we use little data for optimization (as many as 5 samples in several cases).
>
> > In Figure 2, regression performance deteriorates as the number of prompt elements increases, indicated by larger MAE and smaller r-values. Coupling between prompt elements.
>
> While the results for predicting the effects of 2 prompt elements are the best across experiments, results for 4 elements are often better than for 3, indicating no clear trend w.r.t. the number of prompt elements. Moreover, our prompt regression yields $r \approx 0.9$ in most cases, indicating that a linear model fits data well, i.e. there is no significant effect due to coupling between the prompt elements.
>
> > "minimally-prompted model" and baselines
>
> It is an LLM with basic instructions relative to the task (e.g., “You are a hate speech detector”; see Appendix B.1). We use it to see the effects of our prompt library relative to those of instruction tuning. Furthermore, we approximate methods involving constrained brute-force searches (e.g., Prasad et al. (2023)) by reporting 75th percentile and maximum possible metrics from all element combinations, but we can also report results with the method of Shi et al. (2024) in a revision.
>
> > Comparison to random baseline
>
> The random baseline is quite strong when the prompt library is good, as can be seen by comparing results to 75\% percentile and max performance. As noted in Section 5, our method can filter out unhelpful prompt elements and is more interpretable than trying prompts at random. We also reiterate our regression model is an interesting standalone contribution.

---

> > ### Comment · Reviewer_jCnP · 2024-06-03
> > **Response to Authors' Rebuttal**
> >
> > Thank you for addressing my concerns. Your responses clarified the motivation for the prompt library search problem and demonstrated the generalizability of PEPR prompts across related tasks. You also provided useful information on the scale of data used for optimization and the rationale behind the "minimally-prompted model." However, some issues remain unresolved, particularly regarding the completeness of comparisons to advanced methods. While I appreciate the effort and the improvements in understanding, the responses still leave some questions about methodological details. Therefore, I have decided to increase the score from 4 (reject) to a borderline accept, reflecting the clarifications provided and the potential of your approach. Further rigorous validation would enhance the confidence in the findings.

---

> > > ### Author Response · Authors · 2024-06-05
> > >
> > > Thank you for your feedback and for increasing the score. We will incorporate the clarifications into the revised draft. We will also add a comparison to Shi et al. (2024) (a recent method studying a related problem setting, but limited to smaller prompt libraries) to strengthen the empirical validation as you suggested.

---

### Official Review · Reviewer_cEYe · 2024-05-11

**Rating:** 8
**Confidence:** 4
**Ethics Flag:** 1

**Summary:**

The paper introduces a framework, Prompt Exploration with Prompt Regression (PEPR), which aims to optimize the selection of prompt elements for language models. It claims that unlike prior methods, PEPR can predict the output effects of various prompt combinations without exhaustive testing. The authors present a methodology involving building a library of prompt elements, assigning weights through regression analysis, and selecting the optimal combination based on these weights. The paper asserts that its approach, tested on various datasets and model sizes, improves prompt selection efficiency and effectiveness. It suggests that PEPR could significantly streamline the process of prompt engineering, although it also calls for further research to refine and expand the method.

Overall, the paper is well written and introduces an approach for which the COLM conference is arguably the perfect venue. Hence, we believe it should be accepted.

**Questions To Authors:**

1. Can you shed further light on what you mean by "solving (3.3) does not require knowledge of the reference (correct) y for the inputs (e.g., when there is a finite set of meaningful generations (such as classification or multiple-choice QA), one can simply plug every possible y for every input xi when fitting the regression coefficients)". In a quick reading it might sound like the PEPR method only supports classification usecases but that is patently not the case when considering the rest of the paper.

2. Would it be possible to quantify the level of interaction between prompt elements when their effects on language model outputs are not independent and make use of this information in the prompt selection step?

**Reasons To Accept:**

- PEPR provides a method to predict the outcomes of combined prompt elements without testing each combination, addressing the inefficiency of brute-force methods by providing a more principled method of building prompts from a selection of templates.
- The framework integrates regression analysis to assign quantitative impacts to individual prompt components, allowing for systematic optimization.
- PEPR utilizes both reference text generations and preference data, offering flexibility in aligning language model outputs with human judgments or desired behaviors.
- The method is validated across multiple open-source language models of the Llama2 family and four diverse tasks, showing its adaptability and broad potential for future utilization.
- PEPR introduces a novel approach to prompt library search, a previously underexplored area in language model research.

**Reasons To Reject:**

- The paper does not fully address the scalability of PEPR with very large prompt libraries or complex prompt structures, which could limit its practicality in extensive applications.
- The methodology may not account adequately for the nuanced interactions between prompt elements beyond their individual contributions, potentially oversimplifying complex linguistic phenomena.
- Further, the methodology relies heavily on the initial quality and composition of the prompt library, potentially limiting its effectiveness if the library is not well-constructed.
- The experiments lack a comparison against a wider range of existing prompt engineering methods, which could help in understanding PEPR's relative performance.

All in all, these reasons are relatively superficial and do not warrant the rejection of the paper. Moreover, the authors address most of them in the manuscript to a great extent.

---

> ### Author Rebuttal · Authors · 2024-05-29
>
> We thank you for your comments on our work and your very positive review. Please see our responses below:
>
> > Can you shed further light on what you mean by "solving (3.3) does not require knowledge of the reference (correct) y for the inputs (e.g., when there is a finite set of meaningful generations (such as classification or multiple-choice QA), one can simply plug every possible y for every input xi when fitting the regression coefficients)". In a quick reading it might sound like the PEPR method only supports classification usecases but that is patently not the case when considering the rest of the paper.
>
> We mean that of a possible set of outputs, we do not need to know which one is correct in order to perform regression. This is because our regression approach models log-probabilities (and corresponding differences) of outputs themselves rather than assigning probabilities of correctness. In contrast, our proposed selection method *does* require knowledge of which outputs are correct in order to derive a performant prompt relative to correct outputs (see Table 1). Note that in using terms like “correct” and “possible set of outputs”, we do not mean to imply that our methods are only suited for classification (as you have deduced) but rather that even a given set of open-ended responses has associated qualities of correctness and desirability that are needed for selection and not for regression.
>
> > Would it be possible to quantify the level of interaction between prompt elements when their effects on language model outputs are not independent and make use of this information in the prompt selection step?
>
> This is an interesting idea. Though we do not explore such an approach, it may be possible to detect dependence between prompt elements whenever our regression model is repeatedly or on average very inaccurate (relative to ground-truth log-probabilities and differences) for a given prompt element combination. In doing so, perhaps coefficients could be tweaked for specific combinations of elements such that selection is more accurate. More complicated models like linear regressions with interaction terms and factorization machines could account for pairwise interactions and increase effectiveness (perhaps at the risk of reducing interpretability). At any rate, this line of reasoning could be a fruitful future direction of work.

---

> > ### Comment · Reviewer_cEYe · 2024-06-05
> >
> > Thank you for your responses. I would like to keep my original scores.

---

### Official Review · Reviewer_t5Qb · 2024-05-19

**Rating:** 8
**Confidence:** 3
**Ethics Flag:** 1

**Summary:**

This work studies the prompt library search problem where the model needs to predict prompt combinations in a library that can lead to optimal performance. This work further proposes a method including three steps: (1) building a prompt library; (2) generating weight for each element; (3) prompt selection based on weights.

**Reasons To Accept:**

- The problem formulation of prompt library search is novel and interesting but could benefit from more explanation of its motivation and practical usage.

- This work provides well-formulated mathematical definitions and formulae for prompt regression and prompt selection.

- This framework enables quantitative analysis of individual prompt component's impact in regression analysis.

- The experiments are conducted across a wide range of language models.

**Reasons To Reject:**

- There is a lack of comparison with existing prompt engineering methods. The existing comparison with a random baseline is not enough to show the advantage of the proposed PEPR method.

- The performance of this method highly relies on the quality of the prompt library.

---

> ### Author Rebuttal · Authors · 2024-05-29
>
> Thank you for your feedback and your highly positive opinion of our work. Please see our responses to your comments below:
>
> >  There is a lack of comparison with existing prompt engineering methods. The existing comparison with a random baseline is not enough to show the advantage of the proposed PEPR method.
>
> While we concede that we primarily compare results of PEPR with a random baseline, we also report 75th percentile performance metrics (relative to results from all possible prompt element combinations) in Table 2 and similar tables in our Appendix. In doing so, we approximate the results of brute-force searches with a fixed computation budget, which in turn are explored in other parts of the prompt engineering literature (e.g., GrIPS by Prasad et al. (2023)). We can also opt to include the performance results of the bandit algorithm introduced by Shi et al. (2024) in a revision of our paper.
>
> > The performance of this method highly relies on the quality of the prompt library.
>
> Other related prompt engineering work (like that of Shi et al. (2024)) also assumes a prompt library or similar is used as a starting point for iterating on prompts. Therefore, though our method is dependent on the prompt library’s quality, this is also a constraint of other prompt engineering methods. Furthermore, other works consider relatively few prompt candidates (up to 100) whereas we are able to consider more (around 1000 due to possible combinations of prompt elements) because we account for the compositional structure of prompts.

---

> > ### Comment · Reviewer_t5Qb · 2024-06-07
> >
> > Thank you for your responses. I will keep my original scores.

---

### Comment · Area_Chair_vSns · 2024-06-02
**Discussion period is now open**

Hi reviewers, please take a look at the author's rebuttals and the other reviews for this paper!

If the rebuttals addressed your concerns, please let the authors know about this and update your review. If not, please continue to engage with the authors and the other reviews in the discussion forum.

There is significant disagreement among the reviewers here.

jCnP, you are the most negative about this paper, particularly with concerns about the limitations of using a prompt library constructed a priori, and a lack of experiments comparing with prior methods, plus a lack of significant improvement over baseline methods. Did the authors' response address these concerns?

t5Qb and cEye, you are both quite positive about the paper. Would you like to argue for its acceptance?

---

### Decision · Program_Chairs · 2024-07-10

**Decision:**

Accept

**Comment:**

This paper explores the problem of automatically identifying suitable prompts. This work takes advantage of similarities between prompt variations and their performance, and proposes a method called Prompt Exploration with Prompt Regression that predicts performance of prompt combinations given performance of component variations.

Experiments cover a wide range of LMs and tasks. Reviewers note that there is not thorough comparison with existing prompt engineering methods (although authors suggest they will include more baselines in a final version of the paper), and also that the proposed method is constrained by the quality of the underlying prompt library. While the proposed approach doesn't always outperform a random baseline, the authors argue that prompt regression facilitates interpretability by showing the influence of prompt components on the overall prompt.